# Conformal Prediction for Hierarchical Data

## Abstract

We consider conformal prediction of multivariate data series, which consists of outputting prediction regions based on empirical quantiles of point-estimate errors. We actually consider hierarchical multivariate data series, for which some components are linear combinations of others. The intuition is that the hierarchical structure may be leveraged to improve the prediction regions in terms of their sizes for given coverage levels. We implement this intuition by including a projection step (also called reconciliation step) in the split conformal prediction [SCP] procedure and prove that the resulting prediction regions are indeed globally smaller than without the projection step. The associated strategies and their analyses rely on the literatures of both SCP and forecast reconciliation. We also illustrate the theoretical findings, both on artificial and on real data.

## 1. Introduction

This article combines two post-hoc procedures (two procedures that are applied after initial forecasts were computed): conformal prediction and forecast reconciliation for hierarchical data, both in a regression setting. Hierarchical data refers to multivariate observations abiding by some linear structure such that some components (referred to as in the higher levels of the hierarchy) are given by linear combinations of a subset of the components (referred to as the lower level of the hierarchy).

**Multivariate conformation prediction.** Conformal prediction is a general approach to output prediction sets, based on finite samples and on any underlying forecasting method, under mild assumptions—typically, exchangeability of data. It was first made formal by Vovk et al. (2005) and gained attention since the work by Lei et al. (2018). We are interested in the multivariate (also called multitarget) extensions

of conformal predictions, discussed by Johnstone & Cox (2021), Messoudi et al. (2021; 2022), and Feldman et al. (2023); see Appendix E for more details. These works deal with prediction regions providing joint coverage guarantees, while we will rather be interested in component-wise coverage guarantees.

**Forecast reconciliation.** Forecast reconciliation is about taking into account the hierarchical structure of the multivariate data to improve forecasts. The intuition guiding this approach is that observations at the higher levels of the hierarchy are often easier to forecast, and that these forecasts can be leveraged to improve the forecasts at the lower levels of the hierarchy. Conversely, valuable local information from the lower levels can be leveraged to improve forecasts at the higher ones. We mainly focus on a series of works (Hyndman et al., 2011, Wickramasuriya et al., 2019, Panagiotelis et al., 2021) that approach reconciliation by projections onto the subspace of so-called coherent forecasts; see Appendix D for more details.

**Contributions and challenges.** This article combines (for the first time) both theories, to provide improved conformal predictions in the case of hierarchical data. The benchmark procedure consists of the split conformal prediction procedure (Lei et al., 2018) run on signed non-conformity scores (as in Linusson et al., 2014) and in a component-wise fashion. The improved version only differs from this benchmark through the application of a projection to the regression function learned on the train set—in the spirit of forecast reconciliation. However, the challenge, and our contributions, lie in showing that for a given coverage level, the resulting prediction regions indeed leverage the hierarchical structure of the data and are more efficient, i.e., smaller in some sense to be made precise. Actually (and only because we are interested in component-wise coverage, see Appendix E.2), the main difficulty was to state and control the corresponding criterion, referred to as (⋆⋆) in the sequel. To do so, we show that one can resort to some known trace inequalities of the literature of forecast reconciliation, and also introduce some new such trace inequalities.

A more precise description of the challenges overcome may be found after the sketch of the proof of Theorem 3.7 in Section 3.2 (see also the specific discussions in Appendices E.2 and D.2).

[1]Anonymous Institution, Anonymous City, Anonymous Region, Anonymous Country. Correspondence to: Anonymous Author <anon.email@domain.com>.

Preliminary work. Under review by the International Conference on Machine Learning (ICML). Do not distribute.

**Unrelated references.** The terminology "hierarchical" also appears in Lee et al. (2023), Dunn et al. (2023) and Duchi et al. (2024) in the contexts of predictive inference or conformal prediction. However, this line of research is about a completely different setting (data coming from different sources) and bears no relationship to the hierarchical structure we consider. Actually, Duchi et al. (2024) rather use the terminology "multi-environment" in that context to avoid potential confusions.

**Outline.** In Section 2, we formally state the setting considered, the objectives targeted, and the methodology followed. The objectives consist of a component-wise coverage objective (⋆) and of an efficiency (small-length) objective (⋆⋆). The methodology consists of taking the split conformal procedure (run component-wise with signed non-conformity scores) as a benchmark, and discuss how to improve it by adding a reconciliation step through projections. Section 3 then states the theoretical results achieved: the coverage guarantees in Theorem 3.2, and efficiency results (weak and practical version in Theorem 3.7, strong and oracle version in Theorem 3.10). We only provide a sketch of the proof of Theorem 3.7 (to give an idea of how we connected the tools of conformal prediction and of forecast reconciliation), and defer full proofs of all these results in Appendices A–Appendix B–Appendix C. Finally, Section 4 illustrates the theoretical finds both on artificial and on real data, with full details of the simulations to be found in Appendix F. The real data concern the charging of electric vehicles. We recall that Appendices D and E discuss the literature of forecast reconciliation and multivariate split conformal prediction, respectively, and review their classic results.

**Notation.** For an integer $n \geqslant 1$, let $[n] = \{1, \ldots, n\}$. For a real number $x \geqslant 0$, we let $\lfloor x \rfloor$ and $\lceil x \rceil$ denote the lower and upper integer parts, respectively. For a vector $\boldsymbol{u} \in \mathbb{R}^m$ and $n \leqslant m$, let $\boldsymbol{u}_{1:n} = (u_1, \ldots, u_n)^\top$ be the vector of the first $n$ components of $\boldsymbol{u}$. The null vector of $\mathbb{R}^m$ is denoted by $\boldsymbol{0} = (0, \ldots, 0)^\top$. We let $\operatorname{diag}(\boldsymbol{w})$ denote the $m \times m$ diagonal matrix with diagonal elements given by $\boldsymbol{w} \in \mathbb{R}^m$. We denote by $\operatorname{Id}_m$ the $m \times m$ identity matrix. The trace of square matrix $M$ is denoted by $\operatorname{Tr}(M)$.

## 2. Objectives and methodology

**Setting.** We consider a multivariate regression problem of observations $\boldsymbol{y} \in \mathbb{R}^m$, where $m \geqslant 3$, based on features $\boldsymbol{x} \in \mathbb{R}^d$, where $d \geqslant 1$. The observations enjoy some hierarchical structure: some of their components (henceforth referred to as aggregated levels) are given by sums over subsets of other components (henceforth referred to as the most disaggregated levels). More formally, up to reordering the components of $\boldsymbol{y}$, there exist $2 \leqslant n < m$ and a $m \times n$

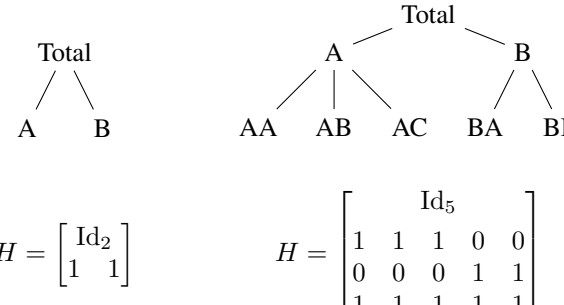

$$H = \begin{bmatrix} \operatorname{Id}_2 \\ 1 \quad 1 \end{bmatrix} \qquad H = \begin{bmatrix} \operatorname{Id}_5 \\ 1 \ 1 \ 1 \ 0 \ 0 \\ 0 \ 0 \ 0 \ 1 \ 1 \\ 1 \ 1 \ 1 \ 1 \ 1 \end{bmatrix}$$

*Figure 1.* Two examples of hierarchical structures and their associated structural matrices $H$.

matrix $H$ of the form

$$H = \begin{bmatrix} \operatorname{Id}_n \\ H_{\mathrm{sub}} \end{bmatrix} \quad \text{such that} \quad \boldsymbol{y} = H\boldsymbol{y}_{1:n}, \qquad (1)$$

where $H_{\mathrm{sub}}$ is any $(m-n) \times n$ matrix of real numbers. The matrix $H$ encoding the hierarchical summation constraints is called the structural[1] matrix. Two examples are provided in Figure 1, of the "bottom-up" form (i.e., with matrices $H_{\mathrm{sub}}$ of some specific form, but we recall that we will require no specific assumption on $H_{\mathrm{sub}}$).

**Definition 2.1.** Vectors $\boldsymbol{u} \in \mathbb{R}^m$ satisfying the linear constraints $\boldsymbol{u} = H\boldsymbol{u}_{1:n}$ are called coherent. The subspace $\operatorname{Im}(H)$ of all such vectors is called the coherent subspace.

### 2.1. Objectives

We assume that i.i.d. data $(\boldsymbol{x}_t, \boldsymbol{y}_t)_{1 \leqslant t \leqslant T}$ is available to perform the regression task (we do so for the sake of exposition; Assumption 3.1 will later relax this requirement). We now describe the objective(s) targeted: constructing prediction rectangles with a coverage objective (⋆) and a small-length objective (⋆⋆).

The primary objective is to construct prediction sets based on this $T$–sample, of the form $C_1 \times \ldots \times C_m$, where each $C_i : \boldsymbol{x} \in \mathbb{R}^d \mapsto C_i(\boldsymbol{x})$ is an application taking subsets of $\mathbb{R}$ as values. Consider a new data point $(\boldsymbol{x}_{T+1}, \boldsymbol{y}_{T+1})$ i.i.d. from the $T$–sample. The prediction sets should be such that each component $y_{T+1,i}$ of the observations $\boldsymbol{y}_{T+1}$ be predicted by $C_i(\boldsymbol{x}_{T+1})$ with a coverage level of approximatively $1 - \alpha$:

$$\forall i \in [m], \quad \mathbb{P}\big(y_{T+1,i} \in C_i(\boldsymbol{x}_{T+1})\big) \approx 1 - \alpha, \quad (\star)$$

where the probability $\mathbb{P}$ is with respect to both $(\boldsymbol{x}_{T+1}, \boldsymbol{y}_{T+1})$ and $(\boldsymbol{x}_t, \boldsymbol{y}_t)_{1 \leqslant t \leqslant T}$.

---

[1] In the literature of forecast reconciliation, this matrix is usually denoted by $S$. We rather keep this letter for non-conformity scores. Also, the most disaggregated levels are often the last $n$ components, while we consider the first $n$ components.

This objective is different from the typical objective in other contributions on multivariate conformal prediction (see Messoudi et al., 2021, Messoudi et al., 2022, Feldman et al., 2023), which is about a global (not component-wise) coverage guarantee: output a prediction region $C(\boldsymbol{x}_{T+1})$ such that

$$\mathbb{P}\big(\boldsymbol{y}_{T+1} \in C(\boldsymbol{x}_{T+1})\big) \approx 1 - \alpha \,. \qquad (2)$$

We are interested in the objective $(\star)$ because we want to possibly concentrate on specific nodes within the hierarchy and get individual coverage guarantees for them. For instance, in our real-data application (see Section 4), this could correspond to making sure that each recharging station is well-dimensioned.

In addition, we discuss a simple way of leveraging the approach developed later in this article for the joint-coverage objective (2): see Appendix E.

A secondary objective is to ensure that the prediction sets output are efficient, i.e., are as small as possible. We will be mostly interested in rectangles $C_1(\boldsymbol{x}) \times \ldots \times C_m(\boldsymbol{x})$ based on intervals $C_i(\boldsymbol{x}) = \big[\mu_i(\boldsymbol{x}) + a_i(\boldsymbol{x}),\ \mu_i(\boldsymbol{x}) + b_i(\boldsymbol{x})\big]$, whose respective lengths are denoted by

$$\ell\big(C_i(\boldsymbol{x})\big) = b_i(\boldsymbol{x}) - a_i(\boldsymbol{x}) \,.$$

One quantification of the size of such a rectangle is given by

$$\sum_{i=1}^{m} w_i \, \ell\big(C_i(\boldsymbol{x})\big)^2 \,,$$

where we fix some vector $\boldsymbol{w} = (w_1, \ldots, w_m)^\top$ of positive numbers. The weights $\boldsymbol{w}$ may be used to ponder the components based on their respective importance. The secondary objective then formally corresponds to

$$\text{minimizing} \quad \mathbb{E}\left[\sum_{i=1}^{m} w_i \, \ell\big(C_i(\boldsymbol{x}_{T+1})\big)^2\right], \qquad (\star\star)$$

where, again, the expectation $\mathbb{E}$ is with respect to both $(\boldsymbol{x}_{T+1}, \boldsymbol{y}_{T+1})$ and $(\boldsymbol{x}_t, \boldsymbol{y}_t)_{1 \leqslant t \leqslant T}$.

## 2.2. Hierarchical Split Conformal Prediction

**Positioning.** Lei et al. (2018) introduced a variant of conformal prediction called split conformal prediction [SCP], a procedure based on splitting data between a training set (to learn a regressor function) and a calibration set (to compute estimation errors, a.k.a. residuals or non-conformity scores). This procedure has been extensively studied in the univariate case, and often through considering the absolute values of the non-conformity scores, which leads to centered intervals. We are interested in two extensions of this basic setting: multivariate SCP and signed non-conformity scores.

Signed non-conformity scores were already considered in the univariate case by Linusson et al. (2014). They are handy

in our setting because we consider linear constraints: the signed non-conformity scores $\widehat{\boldsymbol{s}} = \boldsymbol{y} - \widehat{\boldsymbol{y}}$ between coherent observations $\boldsymbol{y}$ and forecasts $\widehat{\boldsymbol{y}}$ are also coherent, while the vector of their absolute values is not coherent in general.

Multivariate SCP was already studied by Messoudi et al. (2021) but with somewhat different objectives: in particular, the design of prediction regions for the vectors $\boldsymbol{y}_{T+1}$, while our objective $(\star\star)$ is about separate prediction intervals for each component. A more-in-depth discussion of the links and differences between the objectives of Messoudi et al. (2021) and $(\star\star)$ may be found in Appendix E.

**Formal description: plain multivariate version.** Formally, the component-wise coverage objectives $(\star)$ may be achieved by a component-wise SCP, as follows. Data splitting corresponds to partitioning $[T]$ into the subsets $\mathcal{D}_{\text{train}}$ and $\mathcal{D}_{\text{calib}}$, with respective cardinalities denoted by $T_{\text{train}}$ and $T_{\text{calib}}$. With pairs $(\boldsymbol{x}_t, \boldsymbol{y}_t)$ indexed by $t \in \mathcal{D}_{\text{train}}$, a regressor function $\widehat{\boldsymbol{\mu}} : \boldsymbol{x} \in \mathbb{R}^d \mapsto \widehat{\boldsymbol{\mu}}(\boldsymbol{x}) \in \mathbb{R}^m$ is built, thanks to some regression algorithm $\mathcal{A}$ provided as input parameter to the SCP procedure. Then, on the calibration set, i.e., for each $t \in \mathcal{D}_{\text{calib}}$, point estimates $\widehat{\boldsymbol{y}}_t = \widehat{\boldsymbol{\mu}}(\boldsymbol{x}_t)$ and signed non-conformity scores (also known as signed estimation errors or signed residuals) $\widehat{\boldsymbol{s}}_t = \boldsymbol{y}_t - \widehat{\boldsymbol{y}}_t$ are computed. The component-wise character of the procedure appears in the third and final step, where prediction intervals are output component by component. Indeed, for each component $i \in [m]$, we order separately the $i$–th components $\widehat{s}_{t,i}$ of the non-conformity scores $\widehat{\boldsymbol{s}}_t$, where $t \in \mathcal{D}_{\text{calib}}$; the ordered values are denoted as follows, by using the standard notation of order statistics:

$$\widehat{s}_{(1),i} \leqslant \ldots \leqslant \widehat{s}_{(T_{\text{calib}}),i} \,.$$

We also define $\widehat{s}_{(0),i} = -\infty$ and $\widehat{s}_{(T_{\text{calib}}+1),i} = +\infty$. Finally, the prediction interval for $y_{T+1,i}$ based on the corresponding features $\boldsymbol{x}_{T+1}$ is

$$\widehat{C}_i(\boldsymbol{x}_{T+1}) = \left[\widehat{\mu}_i(\boldsymbol{x}_{T+1}) + \widehat{s}_{\big(\lfloor (T_{\text{calib}}+1)\alpha/2 \rfloor\big),i} \,,\right.$$
$$\left. \widehat{\mu}_i(\boldsymbol{x}_{T+1}) + \widehat{s}_{\big(\lceil (T_{\text{calib}}+1)(1-\alpha/2) \rceil\big),i}\right],$$

where $1 - \alpha \in (0, 1)$ is the confidence level targeted and where $\widehat{\mu}_i(\boldsymbol{x}_{T+1})$ is the $i$–th component of the point estimate $\widehat{\boldsymbol{\mu}}(\boldsymbol{x}_{T+1})$. The prediction rectangle for $\boldsymbol{y}_{T+1}$ is the Cartesian product of the prediction intervals $\widehat{C}_i(\boldsymbol{x}_{T+1})$.

The thus-defined plain multivariate version of SCP with signed non-conformity scores is summarized in Algorithm 1. We use it as a benchmark and now introduce a generalization of this algorithm taking the hierarchical structure $H$ into account.

---

**Algorithm 1** Plain multivariate SCP with signed scores

---

**Parameters:** confidence level $1 - \alpha$; regression algorithm $\mathcal{A}$; partition of $[T]$ into subsets $\mathcal{D}_{\text{train}}$ and $\mathcal{D}_{\text{calib}}$ of respective cardinalities $T_{\text{train}}$ and $T_{\text{calib}}$

1: Build the regressor $\widehat{\boldsymbol{\mu}}(\cdot) = \mathcal{A}\big((\boldsymbol{x}_t, \boldsymbol{y}_t)_{t \in \mathcal{D}_{\text{train}}}\big)$
2: Denote $\widehat{\boldsymbol{\mu}}(\cdot) = \big(\widehat{\mu}_1(\cdot), \ldots, \widehat{\mu}_m(\cdot)\big)$
3: **for** $t \in \mathcal{D}_{\text{calib}}$ **let** $\widehat{\boldsymbol{y}}_t = \widehat{\boldsymbol{\mu}}(\boldsymbol{x}_t)$ and $\widehat{\boldsymbol{s}}_t = \boldsymbol{y}_t - \widehat{\boldsymbol{y}}_t$
4: **for** each $i \in [m]$ **do**
5:   order the $(\widehat{s}_{t,i})_{t \in \mathcal{D}_{\text{calib}}}$ into $\widehat{s}_{(1),i} \leqslant \ldots \leqslant \widehat{s}_{(T_{\text{calib}}),i}$ and define $\widehat{s}_{(0),i} = -\infty$ and $\widehat{s}_{(T_{\text{calib}}+1),i} = +\infty$
6:   let $\widehat{q}_{\alpha/2}^{(i)} = \widehat{s}_{\big(\lfloor (T_{\text{calib}}+1)\alpha/2 \rfloor\big),i}$
   and $\widehat{q}_{1-\alpha/2}^{(i)} = \widehat{s}_{\big(\lceil (T_{\text{calib}}+1)(1-\alpha/2) \rceil\big),i}$
7:   set $\widehat{C}_i(\cdot) = \Big[\widehat{\mu}_i(\cdot) + \widehat{q}_{\alpha/2}^{(i)},\ \widehat{\mu}_i(\cdot) + \widehat{q}_{1-\alpha/2}^{(i)}\Big]$
8: **return** $\widehat{C}_1(\boldsymbol{x}_{T+1}), \ldots, \widehat{C}_m(\boldsymbol{x}_{T+1})$

---

**Algorithm 2** Hierarchical SCP with signed scores

---

**Parameters:** confidence level $1 - \alpha$; regression algorithm $\mathcal{A}$; partition of $[T]$ into subsets $\mathcal{D}_{\text{train}}$ and $\mathcal{D}_{\text{calib}}$ of respective cardinalities $T_{\text{train}}$ and $T_{\text{calib}}$; matrix $P$

1: Build the regressor $\widehat{\boldsymbol{\mu}}(\cdot) = \mathcal{A}\big((\boldsymbol{x}_t, \boldsymbol{y}_t)_{t \in \mathcal{D}_{\text{train}}}\big)$
2: Denote $P\widehat{\boldsymbol{\mu}}(\cdot) = \widetilde{\boldsymbol{\mu}}(\cdot) = \big(\widetilde{\mu}_1(\cdot), \ldots, \widetilde{\mu}_m(\cdot)\big)$
3: **for** $t \in \mathcal{D}_{\text{calib}}$ **let** $\widetilde{\boldsymbol{y}}_t = \widetilde{\boldsymbol{\mu}}(\boldsymbol{x}_t)$ and $\widetilde{\boldsymbol{s}}_t = \boldsymbol{y}_t - \widetilde{\boldsymbol{y}}_t$
4: **for** each $i \in [m]$ **do**
5:   order the $(\widetilde{s}_{t,i})_{t \in \mathcal{D}_{\text{calib}}}$ into $\widetilde{s}_{(1),i} \leqslant \ldots \leqslant \widetilde{s}_{(T_{\text{calib}}),i}$ and define $\widetilde{s}_{(0),i} = -\infty$ and $\widetilde{s}_{(T_{\text{calib}}+1),i} = +\infty$
6:   let $\widetilde{q}_{\alpha/2}^{(i)} = \widetilde{s}_{\big(\lfloor (T_{\text{calib}}+1)\alpha/2 \rfloor\big),i}$
   and $\widetilde{q}_{1-\alpha/2}^{(i)} = \widetilde{s}_{\big(\lceil (T_{\text{calib}}+1)(1-\alpha/2) \rceil\big),i}$
7:   set $\widetilde{C}_i(\cdot) = \Big[\widetilde{\mu}_i(\cdot) + \widetilde{q}_{\alpha/2}^{(i)},\ \widetilde{\mu}_i(\cdot) + \widetilde{q}_{1-\alpha/2}^{(i)}\Big]$
8: **return** $\widetilde{C}_1(\boldsymbol{x}_{T+1}), \ldots, \widetilde{C}_m(\boldsymbol{x}_{T+1})$

---

**Formal description: hierarchical version of SCP.** The hierarchical version of SCP is stated in Algorithm 2 and only differs from the plain multivariate version stated as Algorithm 1 in line 2, where a projection matrix $P$ onto the coherent subspace $\text{Im}(H)$ should be used: the regressor function considered is $\widetilde{\boldsymbol{\mu}} = P\widehat{\boldsymbol{\mu}}$, instead of simply $\widehat{\boldsymbol{\mu}}$, and thus outputs point estimates that are coherent in the case where $\text{Im}(P) \subseteq \text{Im}(H)$. The rest of the procedure is similar.

Algorithm 1 is a special case of Algorithm 2, for the choice $P = \text{Id}_m$. We however provide two separate statements to clarify the notation: $\widehat{\cdot}$–type quantities are for the plain multivariate version (Algorithm 1), which we use as a benchmark, while $\widetilde{\cdot}$–type quantities refer to the hierarchical version (Algorithm 2) to be used with a projection matrix $P$ onto $\text{Im}(H)$.

**Projection matrices.** As indicated, we will be mostly interested in projection matrices $P$ onto $\text{Im}(H)$ for Algorithm 2. We will most often take them of the form $P_W = H\big(H^\top W H\big)^{-1} H^\top W$, where $W$ is a symmetric definite positive matrix; see Lemma B.5 in Appendix B for a proof that the $P_W$ are indeed projections onto $\text{Im}(H)$, as well as a statement of additional properties that they enjoy.

# 3. Statements of the theoretical results

In conformal prediction, results typically hold in great generality. In particular, we will require no direct assumption on the regression algorithm $\mathcal{A}$, which will be treated as a black-box regression procedure that does not even have to output coherent point estimates (hence the consideration of a projection matrix $P$ in Algorithm 2). No assumption will be required on the data split between $\mathcal{D}_{\text{train}}$ and $\mathcal{D}_{\text{calib}}$, but the following key assumption of i.i.d. behavior will be issued on the data (or only on the non-conformity scores).

**Assumption 3.1.** The non-conformity scores $\widehat{\boldsymbol{s}}_t = \boldsymbol{y}_t - \widehat{\boldsymbol{y}}_t$, for $t \in \mathcal{D}_{\text{calib}} \cup \{T + 1\}$ of the plain multivariate version of SCP (Algorithm 1) are i.i.d. This is in particular the case when data $(\boldsymbol{x}_t, \boldsymbol{y}_t)_{1 \leqslant t \leqslant T+1}$ is i.i.d.

The second part of Assumption 3.1 follows from the fact that $\widehat{\boldsymbol{s}}_t = \boldsymbol{y}_t - \widehat{\boldsymbol{\mu}}(\boldsymbol{x}_t)$, where $\widehat{\boldsymbol{\mu}}$ only depends on the data in $\mathcal{D}_{\text{train}}$ and is therefore independent from the data $(\boldsymbol{x}_t, \boldsymbol{y}_t)$ for $t \in \mathcal{D}_{\text{calib}} \cup \{T + 1\}$.

## 3.1. Coverage objective $(\star)$

The theorem below is related to objective $(\star)$. It of course holds for Algorithm 1, given that it is a special case of Algorithm 2 with $P = \text{Id}_m$. At this stage, no assumption is required on the matrix $P$. The standard proof (together with references to earlier similar proofs) may be found in Appendix A.

**Theorem 3.2** (Coverage). *Fix $\alpha \in (0, 1)$. Algorithm 2, used with any regression algorithm $\mathcal{A}$ and any matrix $P$ such that $PH = H$, ensures that whenever Assumption 3.1 (i.i.d. scores) holds,*

$$\forall i \in [m], \quad \mathbb{P}\big(y_{T+1,i} \in \widetilde{C}_i(\boldsymbol{x}_{T+1})\big) \geqslant 1 - \alpha\,.$$

*In addition, if the non-conformity scores $\big(\widehat{\boldsymbol{s}}_t\big)_{t \in \mathcal{D}_{\text{calib}} \cup \{T+1\}}$ are almost surely distinct, then*

$$\forall i \in [m], \quad \mathbb{P}\big(y_{T+1,i} \in \widetilde{C}_i(\boldsymbol{x}_{T+1})\big) \leqslant 1 - \alpha + \frac{2}{T_{\text{calib}} + 1}\,.$$

## 3.2. Small-length objective $(\star\star)$, weak version

We now move on to the small-length objective $(\star\star)$, and start with a weak version thereof, relying on one fixed weight vector $\boldsymbol{w}$ (see next section for a stronger version).

As mentioned above, we issue no direct assumption on the regression algorithm $\mathcal{A}$. However, we output a mild assumption on the distribution of the non-conformity scores, which may be seen as un indirect assumption on $\mathcal{A}$. This assumption falls within the model-agnostic gist of conformal prediction and is standard in the literature of probabilistic forecast reconciliation (see, e.g., Panagiotelis et al., 2023 or Wickramasuriya, 2024). It states that non-conformity scores follow a distribution, called an elliptical distribution, derived from a spherical distribution; for more details, see Appendix B (which in turns refers to Kollo & von Rosen, 2005, Section 2.3).

**Definition 3.3.** A random vector $\boldsymbol{z}$ follows a spherical distribution over $\mathbb{R}^k$ if $\boldsymbol{z}$ and $\Gamma \boldsymbol{z}$ have the same distribution for all $k \times k$ orthogonal matrices $\Gamma$.

**Definition 3.4.** An elliptical distribution over $\mathbb{R}^m$ is of the form $\boldsymbol{c} + M\boldsymbol{z}$, for a deterministic vector $\boldsymbol{c} \in \mathbb{R}^m$, a $m \times k$ matrix $M$ such that $MM^\top$ has rank $k$, and a random vector $\boldsymbol{z}$ following a spherical distribution over $\mathbb{R}^k$.

A given spherical distribution thus generates a family $\mathcal{F}$ of elliptical distributions enjoying a stability property through linear transformations.

**Example 3.5.** The simplest example of elliptical distributions consists of multivariate normal distributions (which are light-tailed distributions). Other examples include multivariate $t$–distributions and symmetric multivariate Laplace distributions (both heavy tailed) and the uniform distribution on an ellipse (no tail).

We are now ready to state the key assumption used to establish in Theorem 3.7 that the hierarchical version of SCP performs better than its plain multivariate version.

**Assumption 3.6.** The (i.i.d.) non-conformity scores $\widehat{\boldsymbol{s}}_t = \boldsymbol{y}_t - \widehat{\boldsymbol{y}}_t$, for $t \in \mathcal{D}_{\text{calib}}$, of the plain multivariate version of SCP (Algorithm 1) follow some elliptical distribution (whose parameters are unknown to the learner).

**Theorem 3.7.** *Let $\boldsymbol{w} \in \mathbb{R}^m$ be a vector of positive numbers. Under Assumptions 3.1 and 3.6 (i.i.d. scores with elliptical distribution), the hierarchical version of SCP (Algorithm 2) run with $P = P_{\boldsymbol{w}}$, where*

$$P_{\boldsymbol{w}} \stackrel{\text{def}}{=} H\big(H^\top \operatorname{diag}(\boldsymbol{w})H\big)^{-1} H^\top \operatorname{diag}(\boldsymbol{w}), \qquad (3)$$

*provides prediction rectangles that are more efficient than the ones output by the plain multivariate version of SCP (Algorithm 1) in the following sense:*

$$\mathbb{E}\left[\sum_{i=1}^{m} w_i \, \ell\big(\widetilde{C}_i(\boldsymbol{x}_{T+1})\big)^2\right] \leqslant \mathbb{E}\left[\sum_{i=1}^{m} w_i \, \ell\big(\widehat{C}_i(\boldsymbol{x}_{T+1})\big)^2\right].$$

*Sketch of proof.* The centered scores $\widetilde{\boldsymbol{s}}_t - \mathbb{E}\big[\widetilde{\boldsymbol{s}}_t\big]$ are i.i.d. according to a centered elliptical distribution as $t \in \mathcal{D}_{\text{calib}}$.

Thus, their $i$–th components have the same distribution $\nu$ up to a scaling factor denoted by $\sqrt{\gamma_i}$. Let $(v_t)_{t \in \mathcal{D}_{\text{calib}}}$ be i.i.d. variables distributed according to $\nu$, consider their order statistics $v_{(1)} \leqslant \ldots \leqslant v_{(T_{\text{calib}})}$, and set

$$L_\alpha = v_{\left(\lceil (T_{\text{calib}}+1)(1-\alpha/2)\rceil\right)} - v_{\left(\lfloor (T_{\text{calib}}+1)\alpha/2\rfloor\right)}.$$

We thus have that $\ell\big(\widetilde{C}_i(\boldsymbol{x}_{T+1})\big)$ has the same distribution as $\sqrt{\gamma_i}\, L_\alpha$, for each $i \in [m]$. Therefore,

$$\mathbb{E}\left[\sum_{i=1}^{m} w_i \, \ell\big(\widetilde{C}_i(\boldsymbol{x}_{T+1})\big)^2\right] = \mathbb{E}\big[L_\alpha^2\big] \sum_{i \in [m]} w_i \gamma_i. \qquad (4)$$

It may be shown that $\gamma_i$ is the $(i,i)$–th element of a matrix of the form $P_{\boldsymbol{w}} \Gamma P_{\boldsymbol{w}}^\top$, where $\Gamma$ is symmetric positive semi-definite. A similar result holds for the $\widehat{C}_i$, with scaling factors given by the diagonal elements of $\Gamma$. It thus suffices to show that

$$\sum_{i \in [m]} w_i \Gamma_{i,i} \geqslant \sum_{i \in [m]} w_i \big(P_{\boldsymbol{w}} \Gamma P_{\boldsymbol{w}}^\top\big)_{i,i}, \qquad (5)$$

i.e., that $\operatorname{Tr}\big(\operatorname{diag}(\boldsymbol{w})\,\Gamma\big) \geqslant \operatorname{Tr}\big(\operatorname{diag}(\boldsymbol{w})\,P_{\boldsymbol{w}}\Gamma P_{\boldsymbol{w}}^\top\big)$. The latter result essentially follows from taking expectations of a Pythagorean inequality, and is a result of our own, though inspired by the literature of forecast reconciliation.

The complete proof may be found in Appendix B, including a justification of why $P_{\boldsymbol{w}}$ is well defined. $\qquad\square$

**Some words on the challenges overcome.** We sketched above how we connected the tools of conformal prediction and of forecast reconciliation.

The difficulty mostly lied in the component-wise approach, imposed by the component-wise coverage guarantees ($\star$) targeted. Indeed, Appendix E.2 explains in detail why efficiency results are straightforward when some joint coverage is targeted.

More precisely (and as detailed in Appendix D.2), the main blocking point was to relate the minimization of squared lengths (4) to what may be rephrased as some trace minimization (5). Such relationships are classic in the literature of forecast reconciliation, but they rely on assumptions of unbiasedness (i.e., of centered non-conformity scores, which we are not ready to consider). However, by resorting to signed scores, the expectations of the scores cancel out: the distribution of $L_\alpha$ is stable when the $(v_t)_{t \in \mathcal{D}_{\text{calib}}}$ are translated.

### 3.3. Objective ($\star\star$): stronger but oracle version

We resort to tools from the theory of forecast reconciliation to improve the results of Theorem 3.7 and have them hold simultaneously for all possible positive weight vectors $\boldsymbol{w}$.

However, this improvement is only for an oracle strategy relying on a projection matrix $P_{\Sigma^{-1}}$ depending on the covariance matrix $\Sigma$ of the scores (unknown to the learner). This is why we state next (see Algorithm 3) a "practical" implementation of this oracle, which adds an estimation step for $\Sigma$ to the hierarchical SCP procedure stated as Algorithm 2.

**Assumption 3.8.** The (i.i.d.) non-conformity scores $\widehat{s}_t = y_t - \widehat{y}_t$, for $t \in \mathcal{D}_{\text{calib}}$, of the plain multivariate version of SCP (Algorithm 1) have a bounded second-order moment. We denote by $\Sigma$ their (positive definite) covariance matrix.

The projection $P_{\Sigma^{-1}}$ to be considered in Theorem 3.10 is the one used in the Minimum-Trace projection in forecast reconciliation (Wickramasuriya et al., 2019). The name comes from the following optimality result, for which we provide an elementary proof in Appendix C (this proof may be considered in itself as a result of interest). Let

$$P_{\Sigma^{-1}} \stackrel{\text{def}}{=} H\big(H^\top \Sigma^{-1} H\big)^{-1} H^\top \Sigma^{-1}. \qquad (6)$$

**Lemma 3.9** (Minimum-Trace projection). *Let $W$ and $\Sigma$ be two symmetric $m \times m$ matrices, where $W$ positive semi-definite and $\Sigma$ is positive definite. Then, for all projection matrices $P$ onto $\text{Im}(H)$,*

$$\text{Tr}\big(W P_{\Sigma^{-1}} \Sigma P_{\Sigma^{-1}}^\top\big) \leqslant \text{Tr}\big(W P \Sigma P^\top\big).$$

We now state our main results, Theorem 3.10 and Corollary 3.11. Their proofs may be found in Appendix C and consist of direct adaptations of the proof of Theorem 3.7, together with an application of Lemma 3.9.

To do so, we denote by $\widetilde{C}_1^\star(\boldsymbol{x}_{T+1}) \times \ldots \times \widetilde{C}_m^\star(\boldsymbol{x}_{T+1})$ the prediction rectangles output by the hierarchical version of SCP (Algorithm 2) run with $P = P_{\Sigma^{-1}}$ defined in (6), and keep the notation $\widetilde{C}_1(\boldsymbol{x}_{T+1}) \times \ldots \times \widetilde{C}_m(\boldsymbol{x}_{T+1})$ for the prediction rectangles output by the same strategy run with any other choice of a projection matrix.

We obtain the following efficiency result, which is actually stronger than the objective (⋆⋆) stated, as the comparison holds component-wise.

**Theorem 3.10.** *Under Assumptions 3.1, 3.6, and 3.8 (i.i.d. scores with elliptical distribution admitting a second-order moment), the hierarchical version of SCP (Algorithm 2) run with $P = P_{\Sigma^{-1}}$ provides prediction rectangles more efficient than with any other choice of a projection matrix onto $\text{Im}(H)$:*

$$\forall i \in [m], \quad \mathbb{E}\Big[\ell\big(\widetilde{C}_i^\star(\boldsymbol{x}_{T+1})\big)^2\Big] \leqslant \mathbb{E}\Big[\ell\big(\widetilde{C}_i(\boldsymbol{x}_{T+1})\big)^2\Big].$$

**Corollary 3.11.** *In the setting and under the assumptions of Theorem 3.10, we also have*

$$\forall i \in [m], \quad \mathbb{E}\Big[\ell\big(\widetilde{C}_i^\star(\boldsymbol{x}_{T+1})\big)^2\Big] \leqslant \mathbb{E}\Big[\ell\big(\widehat{C}_i(\boldsymbol{x}_{T+1})\big)^2\Big],$$

*where the prediction rectangle $\widehat{C}_1(\boldsymbol{x}_{T+1}) \times \ldots \times \widehat{C}_m(\boldsymbol{x}_{T+1})$ is output by the plain multivariate version of SCP (Algorithm 1).*

The component-wise comparisons stated above in Theorem 3.10 and Corollary 3.11 correspond to inequalities similar to the ones of Theorem 3.7 holding for all positive weight vectors $\boldsymbol{w}$:

$$\forall \boldsymbol{w} \in (0, +\infty)^m, \quad \mathbb{E}\left[\sum_{i=1}^m w_i \, \ell\big(\widetilde{C}_i^\star(\boldsymbol{x}_{T+1})\big)^2\right]$$
$$\leqslant \mathbb{E}\left[\sum_{i=1}^m w_i \, \ell\big(\widetilde{C}_i(\boldsymbol{x}_{T+1})\big)^2\right].$$

(and not just for a single vector $\boldsymbol{w}$ as in Theorem 3.7).

However, the covariance matrix $\Sigma$ is unknown and therefore, running Algorithm 2 with $P_{\Sigma^{-1}}$ is an oracle strategy. We turn it into a practical strategy by adding an estimation step.

### 3.4. Hierarchical SCP with estimated covariance matrix

Theorem 3.10 advocates using Algorithm 2 with $P_{\Sigma^{-1}}$ but the covariance matrix $\Sigma$ of the (unprojected) non-conformity scores is unknown. A natural idea is of course to estimate it: this is why data is now split into three subsets $\mathcal{D}_{\text{train}}$ (train set), $\mathcal{D}_{\text{estim}}$ (estimation set), and $\mathcal{D}_{\text{calib}}$ (calibration set) of respective cardinalities $T_{\text{train}}$, $T_{\text{estim}}$ and $T_{\text{calib}}$. A regression function $\widehat{\boldsymbol{\mu}}$ is still learned on data of $\mathcal{D}_{\text{train}}$, in some black-box way. The covariance matrix of the non-conformity scores built based on $\widehat{\boldsymbol{\mu}}$ is estimated on a fraction of the remaining data, indexed by $\mathcal{D}_{\text{estim}}$. This estimate then determines a projection matrix $P$ to be used to run the final part of Algorithm 2: project scores, rank their components, and deduce the prediction intervals of each component based on the latter rankings.

More precisely, based on the scores $\widehat{\boldsymbol{s}}_t = \boldsymbol{y}_t - \widehat{\boldsymbol{\mu}}(\boldsymbol{x}_t)$, for $t \in \mathcal{D}_{\text{estim}}$, we compute the sample mean $\overline{\boldsymbol{s}}$ and sample covariance matrix $\widehat{\Sigma}$:

$$\overline{\boldsymbol{s}} = \frac{1}{T_{\text{estim}}} \sum_{t \in \mathcal{D}_{\text{estim}}} \widehat{\boldsymbol{s}}_t, \quad \widehat{\Sigma} = \frac{1}{T_{\text{estim}}} \sum_{t \in \mathcal{D}_{\text{estim}}} \big(\widehat{\boldsymbol{s}}_t - \overline{\boldsymbol{s}}\big)\big(\widehat{\boldsymbol{s}}_t - \overline{\boldsymbol{s}}\big)^\top.$$

We assume that $\widehat{\Sigma}$ is symmetric positive definite, which was always the case in our simulations. A projection matrix $P = \mathcal{P}\big(\widehat{\Sigma}\big)$ onto $\text{Im}(H)$ is then computed based on $\widehat{\Sigma}$, through some function $\mathcal{P}$ that operates on symmetric positive definite $m \times m$ matrices. Examples of such functions include

$$\mathcal{P}_{\text{MinT}} : M \longmapsto H\big(H^\top M^{-1} H\big)^{-1} H^\top M^{-1}, \qquad (7)$$

**Algorithm 3** Hierarchical SCP with estimated covariances

**Parameters:** confidence level $1 - \alpha$; regression algorithm $\mathcal{A}$; partition of $[T]$ into three subsets $\mathcal{D}_{\text{train}}$, $\mathcal{D}_{\text{estim}}$, and $\mathcal{D}_{\text{calib}}$ of respective cardinalities $T_{\text{train}}$, $T_{\text{estim}}$ and $T_{\text{calib}}$; function $\mathcal{P}$ operating on $m \times m$ matrices

1: Build the regressor $\widehat{\boldsymbol{\mu}}(\cdot) = \mathcal{A}\big((\boldsymbol{x}_t, \boldsymbol{y}_t)_{t \in \mathcal{D}_{\text{train}}}\big)$

$\{$——*Modifications w.r.t. Algorithm 2 start here*——$\}$

2: **for** $t \in \mathcal{D}_{\text{estim}}$ **let** $\widehat{\boldsymbol{y}}_t = \widehat{\boldsymbol{\mu}}(\boldsymbol{x}_t)$ and $\widehat{\boldsymbol{s}}_t = \boldsymbol{y}_t - \widehat{\boldsymbol{y}}_t$

3: Compute the sample mean of the $(\widehat{\boldsymbol{s}}_t)_{t \in \mathcal{D}_{\text{estim}}}$,

$$\overline{\boldsymbol{s}} = \frac{1}{T_{\text{estim}}} \sum_{t \in \mathcal{D}_{\text{estim}}} \widehat{\boldsymbol{s}}_t, \qquad \text{and their}$$

covariance matrix $\quad \widehat{\Sigma} = \dfrac{1}{T_{\text{estim}}} \sum_{t \in \mathcal{D}_{\text{estim}}} \big(\widehat{\boldsymbol{s}}_t - \overline{\boldsymbol{s}}\big)\big(\widehat{\boldsymbol{s}}_t - \overline{\boldsymbol{s}}\big)^\top$

4: Let $P = \mathcal{P}\big(\widehat{\Sigma}\big)$

$\{$——*Modifications w.r.t. Algorithm 2 stop here*——$\}$

5: Denote $P\widehat{\boldsymbol{\mu}}(\cdot) = \widetilde{\boldsymbol{\mu}}(\cdot) = \big(\widetilde{\mu}_1(\cdot), \ldots, \widetilde{\mu}_m(\cdot)\big)$

6: **for** $t \in \mathcal{D}_{\text{calib}}$ **let** $\widetilde{\boldsymbol{y}}_t = \widetilde{\boldsymbol{\mu}}(\boldsymbol{x}_t)$ and $\widetilde{\boldsymbol{s}}_t = \boldsymbol{y}_t - \widetilde{\boldsymbol{y}}_t$

7: **for** each $i \in [m]$ **do**

8:    order the $(\widetilde{s}_{t,i})_{t \in \mathcal{D}_{\text{calib}}}$ into $\widetilde{s}_{(1),i} \leqslant \ldots \leqslant \widetilde{s}_{(T_{\text{calib}}),i}$ and define $\widetilde{s}_{(0),i} = -\infty$ and $\widetilde{s}_{(T_{\text{calib}}+1),i} = +\infty$

9:    let $\widehat{q}_{\alpha/2}^{(i)} = \widetilde{s}_{\big(\lfloor (T_{\text{calib}}+1)\alpha/2 \rfloor\big),i}$

   and $\widehat{q}_{1-\alpha/2}^{(i)} = \widetilde{s}_{\big(\lceil (T_{\text{calib}}+1)(1-\alpha/2) \rceil\big),i}$

10:   set $\widetilde{C}_i(\cdot) = \Big[\widetilde{\mu}_i(\cdot) + \widehat{q}_{\alpha/2}^{(i)},\ \widetilde{\mu}_i(\cdot) + \widehat{q}_{1-\alpha/2}^{(i)}\Big]$

11: **return** $\widetilde{C}_1(\boldsymbol{x}_{T+1}), \ldots, \widetilde{C}_m(\boldsymbol{x}_{T+1})$

which mimics the expression for $P_{\Sigma^{-1}}$ in Theorem 3.10, corresponding to the Min-Trace projection, hence the $\text{MinT}$ subscript. Other examples are discussed below.

The procedure is summarized in Algorithm 3. The statement of the latter only differs from the one of Algorithm 2 through the additional lines 2–3–4, which consider the estimation set $\mathcal{D}_{\text{estim}}$ and build the projection $P$ based on $\widehat{\Sigma}$.

**Coverage guarantees.** The coverage guarantees of Theorem 3.2 hold also for Algorithm 3 when data $(\boldsymbol{x}_t, \boldsymbol{y}_t)_{1 \leqslant t \leqslant T+1}$ is i.i.d. Indeed, the non-conformity scores $\widetilde{\boldsymbol{s}}_t = \boldsymbol{y}_t - \widetilde{\boldsymbol{y}}_t$, for $t \in \mathcal{D}_{\text{calib}} \cup \{T+1\}$, are then still i.i.d., which is the only property needed in the proof of Theorem 3.2 (located in Appendix A; see the end of its first paragraph). This follows from the fact that these scores are defined based on $\widehat{\boldsymbol{\mu}}$, on $\widehat{\Sigma}$, and on $(\boldsymbol{x}_t, \boldsymbol{y}_t)_{t \in \mathcal{D}_{\text{calib}} \cup \{T+1\}}$, where the latter are independent from $\widehat{\boldsymbol{\mu}}$ and $\widehat{\Sigma}$, which only depend on data of $\mathcal{D}_{\text{train}}$ and $\mathcal{D}_{\text{estim}}$.

**Other examples of projection functions $\mathcal{P}$.** When data is scarce, the estimates $\widehat{\Sigma}$ may be poor, which would cause issues when taking its inverse as in (7). A more robust approach is to consider the vector of the inverses of the

*Table 1.* Summary of the algorithms implemented: nicknames and corresponding formal definitions, through the algorithm number, the required parameter $P$ or $\mathcal{P}$ (if applicable), and the defining equation of the latter. The first five algorithms are practical algorithms, while the sixth algorithm is an oracle.

| Nickname | Algorithm | Parameter | Equation |
|---|---|---|---|
| Direct | Alg. 1 | | |
| OLS | Alg. 2 | $P_{\mathbf{1}}$ | Eq. (3) + (9) |
| MinT | Alg. 3 | $\mathcal{P}_{\text{MinT}}$ | Eq. (7) |
| WLS | Alg. 3 | $\mathcal{P}_{\text{WLS}}$ | Eq. (8) |
| Combi | Alg. 3 | $\mathcal{P}_{\text{Combi}}$ | Eq. (10) |
| Oracle MinT | Alg. 2 | $P_{\Sigma^{-1}}$ | Eq. (6) |

diagonal elements of $\widehat{\Sigma}$ only (i.e., the vector of the inverses of the variances of the components of the scores) and the associated projection matrix $P_{\boldsymbol{w}}$, as in Theorem 3.7. This corresponds to some data-based weighted least squares [WLS], as pointed out by Hyndman et al. (2016):

$$\mathcal{P}_{\text{WLS}} : M \longmapsto H\big(H^\top D_M^{-1} H\big)^{-1} H^\top D_M^{-1}, \qquad (8)$$

where $D_M = \text{diag}\big((M_{i,i})_{i \in [m]}\big)$. In contrast, an ordinary least squares [OLS] approach (considered, for instance, in Hyndman et al., 2011) would be based on the orthogonal projection onto $\text{Im}(H)$, which corresponds to $P_{\mathbf{1}}$ in (3):

$$P_{\mathbf{1}} = H\big(H^\top H\big)^{-1} H^\top, \quad \text{where} \quad \mathbf{1} = (1, \ldots, 1)^\top. \quad (9)$$

Another robust approach could be to use a combination (hence the subscript "Combi") of the $\mathcal{P}$ functions defined above, as suggested by Hollyman et al. (2021), who argued that this combination does not have to be complicated to be efficient. We therefore consider

$$\mathcal{P}_{\text{Combi}} : M \longmapsto \frac{1}{3}\big(P_{\mathbf{1}} + \mathcal{P}_{\text{WLS}}(M) + \mathcal{P}_{\text{MinT}}(M)\big). \quad (10)$$

Lemma B.5 in Appendix B states that the $\mathcal{P}_{\text{MinT}}$, $\mathcal{P}_{\text{WLS}}$, and thus $\mathcal{P}_{\text{Combi}}$ take projection matrices as values.

## 4. Simulations on artificial and real data

We compare the performance achieved by the algorithms presented in this article, as summarized in Table 1: we do so in terms of component-wise coverage levels and of lengths of the prediction intervals. We consider two simulation settings: one on artificially generated data, and one on real data consisting of daily energy demand to charge electric vehicles in Palo Alto, CA. In both cases, the hierarchical structure is the three-level hierarchy (with 8 nodes, of the form 5–2–1) described in the right-hand side of Figure 1.

Full details of the simulation settings (both for artificially generated data and real data) may be found in Appendix F. We only summarize below the main observations.

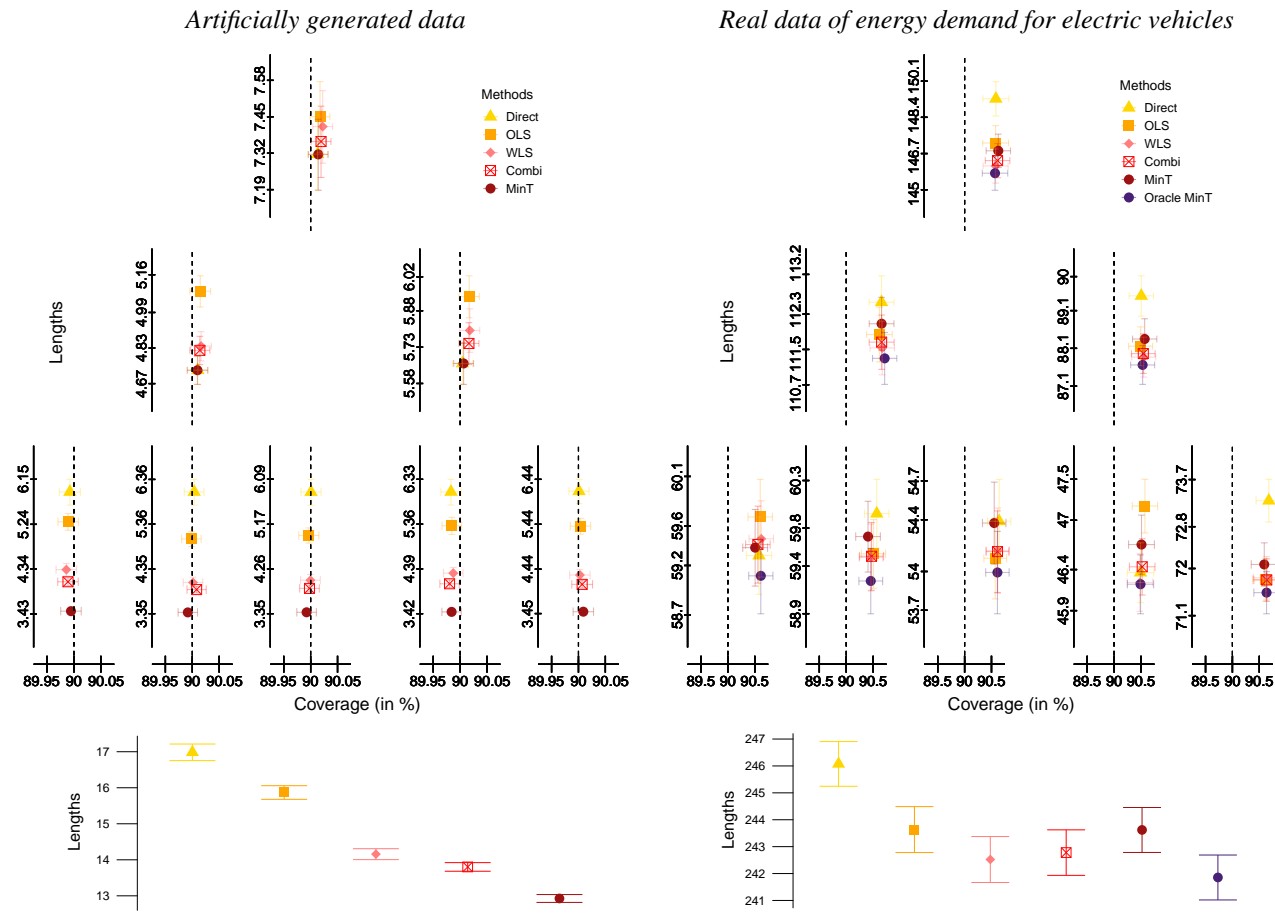

Figure 2. Component-wise coverage levels and prediction-interval lengths (*top graphs*) and total lengths (*bottom graphs*), for artificially generated data (*left graphs*) and real data of energy demand (*right graphs*). Empirical averages are reported, with standard errors (in both the $x$–axis and $y$–axis directions for top graphs). The layout of top graphs follows the tree-representation of the hierarchy.

**Artificially generated data.** We generate $1\,000$ runs of some experimental setting and obtain empirical estimates of the component-wise coverage levels ($x$–axis positions in the top graphs of Figure 2) and root-mean squared lengths ($y$–axis positions of the same graphs). All algorithms obtain component-wise coverage levels close to each other and close to the target value $1 - \alpha = 90\%$. However, differences are clearer in terms of component-wise lengths, with our preferred strategy, MinT, i.e., Algorithm 3 with $\mathcal{P}_{\mathrm{MinT}}$, outperforming all others (as hoped from Theorem 3.10), especially at the lower levels of the hierarchy. Simpler strategies like OLS, i.e., Algorithm 2 with a plain Euclidean projection, provide shorter lengths than Direct, the plain SCP method, but are less efficient than MinT. These differences in lengths are significant as far as the total lengths are concerned: see the bottom graphs of Figure 2, where the confidence intervals on the expected means are all disjoint. The strategies are thus clearly ranked.

**Real data of energy demand.** The data consists of daily charging loads for 5 charging points in two different parking lots (containing respectively 3 and 2 charging points, hence the stated 5–2–1 hierarchy). We consider that the daily data are i.i.d. and generate 360 runs of some experimental setting consisting, in particular, of selecting fractions of the data at random for the train, estimation, and calibration sets. We report empirical averages, as in the case of artificially generated data. All strategies obtain similar component-wise coverage guarantees, close to $90.5\%$. This outcome is in line with Theorem 3.2, which states that the coverage level is actually possibly slightly larger than $1-\alpha$, of a factor of order $1/T_{\mathrm{calib}}$. The sample sizes are much smaller on this use case ($T = 1\,780$ vs. $T = 10^5$ for artificial data), which explains why the slightly larger coverage target appears clearly. Length-wise, the MinT strategy performs relatively poorly, which may be explained by the poor estimation of the covariance matrix (due to lack of data). The WLS and Combi strategies are more robust, as stated Section 3.4. The oracle version of MinT performs the best, as expected (but is an oracle).

## Impact Statement

This paper presents work whose goal is to advance the field of Machine Learning. There are many potential societal consequences of our work, none which we feel must be specifically highlighted here. (The application to real-data of optimizing the charging of electric vehicles is an example of a positive societal consequence, but is only an example among many other possible ones.)

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

## Appendices

The appendices contain the following material: first, detailed proofs of all claims made in the main body, namely,

- A proof of the coverage result (Theorem 3.2), in Appendix A;
- A proof of the efficiency result for fixed weights $\boldsymbol{w}$ (Theorem 3.7), in Appendix B;
- A proof of the component-wise efficiency results (Theorem 3.10 and Corollary 3.11), in Appendix C;

second, detailed reviews (literature review and formal description of classic results) of two topics:

- Forecast reconciliation, in Appendix D;
- Multi-target (multivariate) split conformal prediction, in Appendix E;

third,

- Full details for the simulations on artificial and real data, in Appendix F.

## A. Proof of the coverage result (Theorem 3.2)

**Theorem 3.2** (Coverage). *Fix $\alpha \in (0, 1)$. Algorithm 2, used with any regression algorithm $\mathcal{A}$ and any matrix $P$ such that $PH = H$, ensures that whenever Assumption 3.1 (i.i.d. scores) holds,*

$$\forall i \in [m], \quad \mathbb{P}\big(y_{T+1,i} \in \widetilde{C}_i(\boldsymbol{x}_{T+1})\big) \geqslant 1 - \alpha \,.$$

*In addition, if the non-conformity scores $\big(\widehat{\boldsymbol{s}}_t\big)_{t \in \mathcal{D}_{\mathrm{calib}} \cup \{T+1\}}$ are almost surely distinct, then*

$$\forall i \in [m], \quad \mathbb{P}\big(y_{T+1,i} \in \widetilde{C}_i(\boldsymbol{x}_{T+1})\big) \leqslant 1 - \alpha + \frac{2}{T_{\mathrm{calib}} + 1} \,.$$

The proof below uses a standard methodology in the literature of conformal prediction (see, for instance, Tibshirani et al., 2019, proof of Theorem 1), with rather immediate adaptations due to the multivariate context and to the choice of signed non-conformity scores.

*Proof.* The condition $PH = H$ means that $P$ leaves elements of $\mathrm{Im}(H)$ unchanged. Since observations $\boldsymbol{y}_t$ are coherent, we have, for all $t \in \mathcal{D}_{\mathrm{calib}}$,

$$\widetilde{\boldsymbol{s}}_t \stackrel{\text{def}}{=} \boldsymbol{y}_t - P\widehat{\boldsymbol{y}}_t = P\big(\boldsymbol{y}_t - \widehat{\boldsymbol{y}}_t\big) = P\widehat{\boldsymbol{s}}_t \,.$$

Assumption 3.1 thus entails that the non-conformity scores $\widetilde{\boldsymbol{s}}_t$, where $t \in \mathcal{D}_{\mathrm{calib}} \cup \{T+1\}$, are also i.i.d., thus exchangeable—which is the only property we will use in the rest of this proof.

Fix $i \in [m]$. By definition of $\widetilde{C}_i(\boldsymbol{x}_{T+1})$ and of the score $\widetilde{\boldsymbol{s}}_{T+1} = \boldsymbol{y}_{T+1} - \widetilde{\boldsymbol{\mu}}(\boldsymbol{x}_{T+1})$, the event of interest may be rewritten as

$$\big\{y_{T+1,i} \in \widetilde{C}_i(\boldsymbol{x}_{T+1})\big\} = \Big\{\widetilde{s}_{\big(\lfloor(T_{\mathrm{calib}}+1)\alpha/2\rfloor\big),i} \leqslant \widetilde{s}_{T+1,i} \leqslant \widetilde{s}_{\big(\lceil(T_{\mathrm{calib}}+1)(1-\alpha/2)\rceil\big),i}\Big\} \,. \tag{11}$$

If $\alpha \in (0, 1)$ is so small that $(T_{\mathrm{calib}} + 1)\alpha/2 < 1$, i.e., $\alpha < 2/(T_{\mathrm{calib}} + 1)$, then $\widetilde{s}_{\big(\lfloor(T_{\mathrm{calib}}+1)\alpha/2\rfloor\big),i} = \widetilde{s}_{(0)} = -\infty$ and $\widetilde{s}_{\big(\lceil(T_{\mathrm{calib}}+1)(1-\alpha/2)\rceil\big),i} = \widetilde{s}_{(T_{\mathrm{calib}}+1)} = +\infty$. Thus,

$$\mathbb{P}\big(y_{T+1,i} \in \widetilde{C}_i(\boldsymbol{x}_{T+1})\big) = 1$$

satisfies the claimed statements $\geqslant 1 - \alpha$ and $\leqslant 1 - \alpha + 2/(T_{\mathrm{calib}} + 1)$. Otherwise, $\widetilde{s}_{\big(\lfloor(T_{\mathrm{calib}}+1)\alpha/2\rfloor\big),i}$ and $\widetilde{s}_{\big(\lceil(T_{\mathrm{calib}}+1)(1-\alpha/2)\rceil\big),i}$ correspond respectively to some $\widetilde{s}_{t,i}$ for some $t \in \mathcal{D}_{\mathrm{calib}}$.

We apply arguments of exchangeability in the latter case. The new score $\widetilde{s}_{T+1,i}$ is equally likely to fall into any of the $T_{\mathrm{calib}} + 1$ intervals defined by the $(\widetilde{s}_t)_{t \in \mathcal{D}_{\mathrm{calib}}}$. More formally, by Assumption 3.1, and when scores are almost-surely distinct,

$$\mathbb{P}\big(\widetilde{s}_{T+1,i} < \widetilde{s}_{(1),i}\big) = \mathbb{P}\big(\widetilde{s}_{T+1,i} > \widetilde{s}_{(T_{\mathrm{calib}}),i}\big) = \frac{1}{T_{\mathrm{calib}} + 1}$$

and $\quad \forall k \in [T_{\mathrm{calib}} - 1], \qquad \mathbb{P}\big(\widetilde{s}_{(k),i} < \widetilde{s}_{T+1,i} < \widetilde{s}_{(k+1),i}\big) = \frac{1}{T_{\mathrm{calib}} + 1} \,.$

Therefore, when scores are almost-surely distinct, the event of interest (11) rewrites

$$\left\{ y_{T+1,i} \in \widetilde{C}_i(\boldsymbol{x}_{T+1}) \right\} \overset{\text{a.s.}}{=} \left\{ \widetilde{s}_{\left(\lfloor (T_{\text{calib}}+1)\alpha/2 \rfloor\right),i} < \widetilde{s}_{T+1,i} < \widetilde{s}_{\left(\lceil (T_{\text{calib}}+1)(1-\alpha/2) \rceil\right),i} \right\}$$

and has a probability

$$\mathbb{P}\big(y_{T+1,i} \in \widetilde{C}_i(\boldsymbol{x}_{T+1})\big) = \frac{\lceil (T_{\text{calib}}+1)(1-\alpha/2) \rceil - \lfloor (T_{\text{calib}}+1)\alpha/2 \rfloor}{T_{\text{calib}}+1}$$

$$\leqslant \frac{\big((T_{\text{calib}}+1)(1-\alpha/2)+1\big) - \big((T_{\text{calib}}+1)\alpha/2 - 1\big)}{T_{\text{calib}}+1} = 1 - \alpha + \frac{2}{T_{\text{calib}}+1}\,,$$

as claimed.

We now prove that $\mathbb{P}\big(y_{T+1,i} \in \widetilde{C}_i(\boldsymbol{x}_{T+1})\big) \geqslant 1-\alpha$ whether or not scores are almost-surely distinct. To do so, we show below that

$$\forall k \in [T_{\text{calib}}], \qquad \mathbb{P}\big(\widetilde{s}_{T+1,i} \leqslant \widetilde{s}_{(k),i}\big) \geqslant \frac{k}{T_{\text{calib}}+1} \qquad \text{and} \qquad \mathbb{P}\big(\widetilde{s}_{T+1,i} < \widetilde{s}_{(k),i}\big) \leqslant \frac{k}{T_{\text{calib}}+1}\,, \tag{12}$$

so that, given the rewriting (11), we will end up with

$$\mathbb{P}\big(y_{T+1,i} \in \widetilde{C}_i(\boldsymbol{x}_{T+1})\big) \geqslant \frac{\lceil (T_{\text{calib}}+1)(1-\alpha/2) \rceil - \lfloor (T_{\text{calib}}+1)\alpha/2 \rfloor}{T_{\text{calib}}+1} \geqslant \frac{(T_{\text{calib}}+1)(1-\alpha/2) - (T_{\text{calib}}+1)\alpha/2}{T_{\text{calib}}+1} = 1-\alpha\,.$$

It only remains to show (12). The event $\big\{\widetilde{s}_{T+1,i} \leqslant \widetilde{s}_{(k),i}\big\}$ is exactly the fact that $\widetilde{s}_{T+1,i}$ is among the $k$ smallest elements of the $(\widetilde{s}_t)_{t \in \mathcal{D}_{\text{calib}} \cup \{T+1\}}$. By exchangeability, the probability of the latter event is at least $k/(T_{\text{calib}}+1)$; it may be larger if several scores take the same value as the $k$–th smallest value. Similarly, the event $\big\{\widetilde{s}_{T+1,i} < \widetilde{s}_{(k),i}\big\}$ is exactly the fact that $\widetilde{s}_{T+1,i}$ is among the $k$ smallest elements of the $(\widetilde{s}_t)_{t \in \mathcal{D}_{\text{calib}} \cup \{T+1\}}$ and that there are no ties at the $k$–th smallest value. Due to the additional no-tie condition, and by exchangeability, the probability of the latter event is at most $k/(T_{\text{calib}}+1)$. $\qquad \square$

## B. Proof of the efficiency result for fixed weights $w$ (Theorem 3.7)

We first state some elementary properties of elliptical distributions.

**Property B.1.** The marginals of a spherical distribution are identically distributed. A spherical distribution with a first-order moment is centered: $\mathbb{E}[\boldsymbol{z}] = \boldsymbol{0}$. A spherical distribution with a second-order moment has a covariance matrix proportional to the identity: there exists $\sigma^2 \in [0, +\infty)$ such that $\mathbb{E}\big[\boldsymbol{z}\boldsymbol{z}^\top\big] = \sigma^2\,\text{Id}_k$.

*Proof.* The first property is proved by considering permutation matrices $\Gamma$. The second property holds because $\boldsymbol{u} = \boldsymbol{0}$ is the only vector $\boldsymbol{u} \in \mathbb{R}^k$ such that $\Gamma \boldsymbol{u} = \boldsymbol{u}$ for all orthogonal matrices (first consider permutation matrices to get that all components of $\boldsymbol{u}$ are equal). For the third property, denote by $\Sigma$ the covariance matrix of $\boldsymbol{z}$. Since it is symmetric (positive semi-definite), there exists an orthogonal matrix $\Gamma$ and a vector $\boldsymbol{\lambda} \in \mathbb{R}^k$ (with non-negative elements) such that $\Gamma \Sigma \Gamma = \text{diag}(\boldsymbol{\lambda})$. Now, $\Gamma^\top \boldsymbol{z}$ has the same distribution as $\boldsymbol{z}$, thus their covariance matrices are equal, which shows that $\Sigma = \text{diag}(\boldsymbol{\lambda})$. As marginals have the same distribution, we finally get $\Sigma = \sigma^2\,\text{Id}_k$ for some $\sigma^2 \in [0, +\infty)$, which is actually positive except if the distribution of $\boldsymbol{z}$ is a Dirac at $\boldsymbol{0}$. $\qquad \square$

A slightly more advanced result provides the form of the characteristic function of an elliptical distribution. Its proof is based on first showing that characteristic functions of spherical distributions are exactly of the form $\boldsymbol{u} \mapsto \phi\big(\boldsymbol{u}^\top \boldsymbol{u}\big)$, which is consistent with the fact that spherical distributions are centered. Actually, it may be seen that $\phi$ is the characteristic function of the common distribution of the marginals of $\boldsymbol{z}$.

**Lemma B.2** (Kollo & von Rosen, 2005, Theorem 2.3.5)**.** *Consider a random variable following an elliptical distribution over $\mathbb{R}^m$, of the form $\boldsymbol{c} + M\boldsymbol{z}$, for a deterministic vector $\boldsymbol{c} \in \mathbb{R}^m$, a $m \times k$ matrix $M$ such that $MM^\top$ has rank $k$, and a random vector $\boldsymbol{z}$ following a spherical distribution over $\mathbb{R}^k$. The characteristic function of $\boldsymbol{c} + M\boldsymbol{z}$ is of the form*

$$\forall \boldsymbol{u} \in \mathbb{R}^m, \qquad \mathbb{E}\Big[\exp\big(\text{i}\boldsymbol{u}^\top(\boldsymbol{c} + M\boldsymbol{z})\big)\Big] = \exp\big(\text{i}\boldsymbol{u}^\top \boldsymbol{c}\big)\,\phi\big(\boldsymbol{u}^\top MM^\top \boldsymbol{u}\big)\,,$$

*for some function $\phi : \mathbb{R} \to \mathbb{C}$ that only depends on the distribution of $\boldsymbol{z}$.*

Lemma B.2 is instrumental in showing that the normalized marginals of (a linear transformation of) an elliptical distribution have comparable univariate distributions (that are homothetical), as stated next.

**Lemma B.3.** *With the setting and the notation of Lemma B.2, let $N$ be any $m \times m$ matrix and consider the random vector $\boldsymbol{s} = N(\boldsymbol{c} + M\boldsymbol{z})$. Let $\Lambda = NMM^{\top}N^{\top}$. There exists a random variable $v$, following a univariate distribution induced by the spherical distribution of $\boldsymbol{z}$, such that*

$$\forall i \in [m], \qquad s_i - \mathbb{E}[s_i] \stackrel{(d)}{=} \sqrt{\Lambda_{i,i}} \, v \,.$$

*Proof.* By Lemma B.2, the characteristic function of $\boldsymbol{s} - \mathbb{E}[\boldsymbol{s}]$ is $\boldsymbol{u} \in \mathbb{R}^m \mapsto \phi(\boldsymbol{u}^{\top}\Lambda\boldsymbol{u})$. Thus, the characteristic function of each $s_i - \mathbb{E}[s_i]$ is $u \in \mathbb{R} \mapsto \phi(\Lambda_{i,i}u^2)$. This shows the stated result, for a random variable $v$ with characteristic function $\phi$. $\qquad\square$

A final preliminary is result justifies that the matrix $P_{\boldsymbol{w}}$ introduced in the statement of Theorem 3.7 is well defined.

**Lemma B.4.** *The matrices $H^{\top}H$ and $H^{\top}WH$ are $n \times n$ symmetric positive definite matrices, where $W$ is itself a $m \times m$ symmetric positive definite matrix. Thus, these matrices are invertible.*

*Proof.* The form (1) of $H$ entails that $H^{\top}H = \mathrm{Id}_n + H_{\mathrm{sub}}^{\top}H_{\mathrm{sub}}$, where $H_{\mathrm{sub}}^{\top}H_{\mathrm{sub}}$ is symmetric positive semi-definite. Thus, $H^{\top}H$ is symmetric positive definite. Given it is symmetric positive definite, the matrix $W$ may be decomposed as $W = N^{\top}N$ for some $n \times n$ invertible matrix $N$. The matrix $H^{\top}WH = (NH)^{\top}NH$ is symmetric positive semi-definite. We show that it is even symmetric positive definite: $\boldsymbol{u}^{\top}(NH)^{\top}NH\boldsymbol{u} = 0$ is equivalent to the standard Euclidean norm of $NH\boldsymbol{u}$ being null, thus to $H\boldsymbol{u} = \boldsymbol{0}$ (as $N$ is invertible); given the form (1) of $H$, we conclude that $\boldsymbol{u}^{\top}(NH)^{\top}NH\boldsymbol{u} = 0$ is equivalent to $\boldsymbol{u} = \boldsymbol{0}$, which is the definition of $H^{\top}WH = (NH)^{\top}NH$ being definite. $\qquad\square$

We are now ready to prove Theorem 3.7, which we restate first.

**Theorem 3.7.** *Let $\boldsymbol{w} \in \mathbb{R}^m$ be a vector of positive numbers. Under Assumptions 3.1 and 3.6 (i.i.d. scores with elliptical distribution), the hierarchical version of SCP (Algorithm 2) run with $P = P_{\boldsymbol{w}}$, where*

$$P_{\boldsymbol{w}} \stackrel{\mathrm{def}}{=} H\big(H^{\top} \mathrm{diag}(\boldsymbol{w})H\big)^{-1}H^{\top}\mathrm{diag}(\boldsymbol{w})\,, \tag{3}$$

*provides prediction rectangles that are more efficient than the ones output by the plain multivariate version of SCP (Algorithm 1) in the following sense:*

$$\mathbb{E}\left[\sum_{i=1}^{m} w_i \, \ell\big(\widetilde{C}_i(\boldsymbol{x}_{T+1})\big)^2\right] \leqslant \mathbb{E}\left[\sum_{i=1}^{m} w_i \, \ell\big(\widehat{C}_i(\boldsymbol{x}_{T+1})\big)^2\right]\,.$$

*Proof.* The matrix $P_{\boldsymbol{w}}$ satisfies $P_{\boldsymbol{w}}H = H$, thus, as in the beginning of the proof of Theorem 3.2, we have that for all $t \in \mathcal{D}_{\mathrm{calib}}$,

$$\widetilde{\boldsymbol{s}}_t \stackrel{\mathrm{def}}{=} \boldsymbol{y}_t - P_{\boldsymbol{w}}\widehat{\boldsymbol{y}}_t = P_{\boldsymbol{w}}\big(\boldsymbol{y}_t - \widehat{\boldsymbol{y}}_t\big) = P_{\boldsymbol{w}}\widehat{\boldsymbol{s}}_t \,.$$

We let, for all $t \in \mathcal{D}_{\mathrm{calib}}$ and $i \in [m]$,

$$\widehat{\xi}_{t,i} = \widehat{s}_{t,i} - \mathbb{E}\big[\widehat{s}_{1,i}\big] \qquad \text{and} \qquad \widetilde{\xi}_{t,i} = \widetilde{s}_{t,i} - \mathbb{E}\big[\widetilde{s}_{1,i}\big] \,.$$

By Assumption 3.1 (i.i.d. scores), for each $i \in [m]$, the univariate random variables $\widehat{\xi}_{t,i}$, where $t \in \mathcal{D}_{\mathrm{calib}}$ are i.i.d.; a similar statement holds for the $\widetilde{\xi}_{t,i}$, where $t \in \mathcal{D}_{\mathrm{calib}}$. By Assumption 3.6 and Lemma B.3, there exist a matrix $\Gamma$ of the form $\Gamma = MM^{\top}$ and a random variable $v$ such that, for each $i \in [m]$,

$$\widehat{\xi}_{t,i} \stackrel{(d)}{=} \sqrt{\Gamma_{i,i}} \, v \qquad \text{and} \qquad \widetilde{\xi}_{t,i} \stackrel{(d)}{=} \sqrt{\Gamma'_{i,i}} \, v \,, \qquad \text{where} \qquad \Gamma' = P_{\boldsymbol{w}}\Gamma P_{\boldsymbol{w}}^{\top} \,.$$

Let $(v_t)_{t \in \mathcal{D}_{\mathrm{calib}}}$ be i.i.d. random variables with the same distribution as $v$. We conclude from the facts above that for each $i \in [m]$,

$$\big(\widehat{s}_{t,i}\big)_{t \in \mathcal{D}_{\mathrm{calib}}} \stackrel{(d)}{=} \left(\mathbb{E}\big[\widehat{s}_{1,i}\big] + \sqrt{\Gamma_{i,i}} \, v_t\right)_{t \in \mathcal{D}_{\mathrm{calib}}} \qquad \text{and} \qquad \big(\widetilde{s}_{t,i}\big)_{t \in \mathcal{D}_{\mathrm{calib}}} \stackrel{(d)}{=} \left(\mathbb{E}\big[\widetilde{s}_{1,i}\big] + \sqrt{\Gamma'_{i,i}} \, v_t\right)_{t \in \mathcal{D}_{\mathrm{calib}}} \,.$$

The same equalities in distributions hold for the corresponding order statistics: for each $i \in [m]$,

$$\left(\widehat{s}_{(t),i}\right)_{1 \leqslant t \leqslant T_{\text{calib}}} \overset{(d)}{=} \left(\mathbb{E}\big[\widehat{s}_{1,i}\big] + \sqrt{\Gamma_{i,i}}\; v_{(t)}\right)_{1 \leqslant t \leqslant T_{\text{calib}}} \qquad \text{and} \qquad \left(\widetilde{s}_{(t),i}\right)_{1 \leqslant t \leqslant T_{\text{calib}}} \overset{(d)}{=} \left(\mathbb{E}\big[\widetilde{s}_{1,i}\big] + \sqrt{\Gamma'_{i,i}}\; v_{(t)}\right)_{1 \leqslant t \leqslant T_{\text{calib}}} .$$

By following the conventions of Section 2.2 and letting $v_{(0)} = -\infty$ and $v_{(T_{\text{calib}}+1)} = +\infty$, we even have these equalities in distribution over vectors indexed by $0 \leqslant t \leqslant T_{\text{calib}} + 1$: for each $i \in [m]$,

$$\left(\widehat{s}_{(t),i}\right)_{0 \leqslant t \leqslant T_{\text{calib}}+1} \overset{(d)}{=} \left(\mathbb{E}\big[\widehat{s}_{1,i}\big] + \sqrt{\Gamma_{i,i}}\; v_{(t)}\right)_{0 \leqslant t \leqslant T_{\text{calib}}+1} \text{ and } \left(\widetilde{s}_{(t),i}\right)_{0 \leqslant t \leqslant T_{\text{calib}}+1} \overset{(d)}{=} \left(\mathbb{E}\big[\widetilde{s}_{1,i}\big] + \sqrt{\Gamma'_{i,i}}\; v_{(t)}\right)_{0 \leqslant t \leqslant T_{\text{calib}}+1} .$$

Now, for each $i \in [m]$, by design of Algorithms 1 and 2, the lengths of the intervals $\widehat{C}_i(\boldsymbol{x}_{T+1})$ and $\widetilde{C}_i(\boldsymbol{x}_{T+1})$ output equals

$$\ell\big(\widehat{C}_i(\boldsymbol{x}_{T+1})\big) = \widehat{s}_{\left(\lceil (T_{\text{calib}}+1)(1-\alpha/2)\rceil\right),i} - \widehat{s}_{\left(\lfloor (T_{\text{calib}}+1)\alpha/2\rfloor\right),i}$$

$$\text{and} \qquad \ell\big(\widetilde{C}_i(\boldsymbol{x}_{T+1})\big) = \widetilde{s}_{\left(\lceil (T_{\text{calib}}+1)(1-\alpha/2)\rceil\right),i} - \widetilde{s}_{\left(\lfloor (T_{\text{calib}}+1)\alpha/2\rfloor\right),i} .$$

Thus, letting $L_\alpha = v_{\left(\lceil (T_{\text{calib}}+1)(1-\alpha/2)\rceil\right)} - v_{\left(\lfloor (T_{\text{calib}}+1)\alpha/2\rfloor\right)}$, where $L_\alpha \geqslant 0$ a.s., we finally proved that for each $i \in [m]$,

$$\ell\big(\widehat{C}_i(\boldsymbol{x}_{T+1})\big) \overset{(d)}{=} \sqrt{\Gamma_{i,i}}\; L_\alpha \qquad \text{and} \qquad \ell\big(\widetilde{C}_i(\boldsymbol{x}_{T+1})\big) \overset{(d)}{=} \sqrt{\Gamma'_{i,i}}\; L_\alpha .$$

We showed so far that

$$\mathbb{E}\left[\sum_{i=1}^m w_i\; \ell\big(\widehat{C}_i(\boldsymbol{x}_{T+1})\big)^2\right] = \left(\sum_{i=1}^m w_i\, \Gamma_{i,i}\right) \mathbb{E}\big[L_\alpha^2\big] \qquad \text{and} \qquad \mathbb{E}\left[\sum_{i=1}^m w_i\; \ell\big(\widetilde{C}_i(\boldsymbol{x}_{T+1})\big)^2\right] = \left(\sum_{i=1}^m w_i\, \Gamma'_{i,i}\right) \mathbb{E}\big[L_\alpha^2\big] ,$$

where $\mathbb{E}\big[L_\alpha^2\big] \geqslant 0$ is possibly infinite (in which case the stated result holds). The proof is concluded in the case $\mathbb{E}\big[L_\alpha^2\big] < +\infty$ by noting that

$$\text{Tr}\big(\text{diag}(\boldsymbol{w})\, \Gamma\big) = \sum_{i=1}^m w_i\, \Gamma_{i,i} \geqslant \sum_{i=1}^m w_i\, \Gamma'_{i,i} = \text{Tr}\big(\text{diag}(\boldsymbol{w})\, \Gamma'\big) = \text{Tr}\big(\text{diag}(\boldsymbol{w})\, P_{\boldsymbol{w}} \Gamma P_{\boldsymbol{w}}^\top\big) ,$$

which is guaranteed by the lemma below with $W = \text{diag}(\boldsymbol{w})$, since $\Gamma = MM^\top$ for some $m \times k$ matrix. $\qquad \square$

The first part of Lemma B.5 is elementary. Its second part is inspired by Panagiotelis et al. (2021, Theorem 3.2), which is a result about using orthogonal projections in the $\|\cdot\|_W$–norm to derive distance-reducing properties, and by trace-minimization results that are classic in the literature of forecast reconciliation (like Lemma 3.9 to be found in Appendix C). We however see this second part as a new result of our own. See Appendix D.1, and in particular, the comments after (15), for more details.

**Lemma B.5.** *Fix a symmetric positive definite matrix $W$ and consider the associated inner product and induced norm*

$$\boldsymbol{u}, \boldsymbol{u}' \in \mathbb{R}^m \longmapsto \langle \boldsymbol{u},\, \boldsymbol{u}'\rangle_W = \sqrt{\boldsymbol{u}^\top W \boldsymbol{u}'} \qquad \text{and} \qquad \boldsymbol{u} \in \mathbb{R}^m \longmapsto \|\boldsymbol{u}\|_W \overset{\text{def}}{=} \sqrt{\boldsymbol{u}^\top W \boldsymbol{u}} .$$

*Then, $P_W \overset{\text{def}}{=} H\big(H^\top W H\big)^{-1} H^\top W$ is the orthogonal projection onto $\text{Im}(H)$ in the $\|\cdot\|_W$–norm.*

*Furthermore, for all $m \times k$ matrices $M$,*

$$0 \leqslant \text{Tr}\big(W P_W M M^\top P_W^\top\big) \leqslant \text{Tr}\big(W M M^\top\big) .$$

*Proof.* First, $P_W$ is indeed a projection onto $\text{Im}(H)$: namely, $P_W P_W = P_W$ and $P_W H = H$. To show that $P_W$ is an orthogonal projection for the $\|\cdot\|_W$–norm, it suffices to note that for all $\boldsymbol{u}, \boldsymbol{u}' \in \mathbb{R}^m$,

$$\langle P_W \boldsymbol{u},\, \boldsymbol{u}'\rangle_W \overset{\text{def}}{=} (P_W \boldsymbol{u})^\top W \boldsymbol{u}' = \boldsymbol{u}^\top W P_W \boldsymbol{u}' \overset{\text{def}}{=} \langle \boldsymbol{u},\, P_W \boldsymbol{u}'\rangle_W ,$$

where we used that $P_W^\top W = W P_W$, given the closed-form expression of $P_W$.

Now, let $\boldsymbol{z}'$ be a standard Gaussian random $k$–vector: $\boldsymbol{z}' \sim \mathcal{N}(\boldsymbol{0}, \mathrm{Id}_k)$. On the one hand, given the orthogonality proved for $P_W$ and by a Pythagorean theorem,

$$\|P_W M \boldsymbol{z}'\|_W^2 \leqslant \|M \boldsymbol{z}'\|_W^2 \quad \text{a.s.} \tag{13}$$

Now, by definition of the $\|\cdot\|_W$–norm and by elementary properties of the trace,

$$
\begin{aligned}
\mathbb{E}\big[\|P_W M \boldsymbol{z}'\|_W^2\big] = \mathbb{E}\big[(P_W M \boldsymbol{z}')^\top W P_W M \boldsymbol{z}'\big] &= \mathbb{E}\Big[\mathrm{Tr}\big(W P_W M \boldsymbol{z}'(P_W M \boldsymbol{z}')^\top\big)\Big] \\
&= \mathrm{Tr}\Big(W P_W M \underbrace{\mathbb{E}\big[\boldsymbol{z}'(\boldsymbol{z}')^\top\big]}_{=\mathrm{Id}_k} M^\top P_W^\top\Big) = \mathrm{Tr}\Big(W P_W M M^\top P_W^\top\Big).
\end{aligned}
$$

Similarly, $\quad \mathbb{E}\big[\|M \boldsymbol{z}'\|_W^2\big] = \mathrm{Tr}\big(W M M^\top\big)$.

The inequality (13) and the two equalities proved above conclude the proof. $\qquad\square$

## C. Proof of the component-wise efficiency results (Theorem 3.10 and Corollary 3.11)

The proof of Theorem 3.10, which we restate below, is based on a key equality established in the proof of Theorem 3.7 and on a result that is central in the theory of forecast reconciliation, namely Lemma 3.9 (re-stated and re-proved at the end of this section).

We recall that we denoted by

$$\widetilde{C}_1^\star(\boldsymbol{x}_{T+1}) \times \ldots \times \widetilde{C}_m^\star(\boldsymbol{x}_{T+1}) \qquad \text{and} \qquad \widetilde{C}_1(\boldsymbol{x}_{T+1}) \times \ldots \times \widetilde{C}_m(\boldsymbol{x}_{T+1})$$

the prediction rectangles output by the hierarchical version of SCP (Algorithm 2) run with $P_{\Sigma^{-1}} = H\big(H^\top \Sigma^{-1} H\big)^{-1} H^\top \Sigma^{-1}$ and any other choice of a projection matrix $P$ onto $\mathrm{Im}(H)$, respectively.

**Theorem 3.10.** *Under Assumptions 3.1, 3.6, and 3.8 (i.i.d. scores with elliptical distribution admitting a second-order moment), the hierarchical version of SCP (Algorithm 2) run with $P = P_{\Sigma^{-1}}$ provides prediction rectangles more efficient than with any other choice of a projection matrix onto $\mathrm{Im}(H)$:*

$$\forall i \in [m], \quad \mathbb{E}\Big[\ell\big(\widetilde{C}_i^\star(\boldsymbol{x}_{T+1})\big)^2\Big] \leqslant \mathbb{E}\Big[\ell\big(\widetilde{C}_i(\boldsymbol{x}_{T+1})\big)^2\Big].$$

*Proof.* The proof of Theorem 3.7 did not rely on the existence of a second-order moment, i.e., of a covariance matrix $\Sigma$ for the distribution of the scores $\widehat{\boldsymbol{s}}_t$. (It did not even rely on the existence of a first-order moment.)

When such a second-order moment exists, we may modify the proof of Theorem 3.7 in the following way, to obtain expected lengths depending on $\Sigma$. We also note that though we wrote the beginning of that proof for a specific projection matrix $P_{\boldsymbol{w}}$ onto $\mathrm{Im}(H)$, it holds for all projection matrices $P$ onto $\mathrm{Im}(H)$, and even for all matrices $P$ such that $PH = H$. Namely, when Algorithm 2 is run with any projection matrix $P$ onto $\mathrm{Im}(H)$,

$$\mathbb{E}\left[\sum_{i=1}^m w_i\, \ell\big(\widetilde{C}_i(\boldsymbol{x}_{T+1})\big)^2\right] = \left(\sum_{i=1}^m w_i\, \Gamma_{i,i}'\right) \mathbb{E}\big[L_\alpha^2\big] = \mathrm{Tr}\big(\mathrm{diag}(\boldsymbol{w})\, P\Gamma P^\top\big)\, \mathbb{E}\big[L_\alpha^2\big],$$

where $\Gamma = M M^\top$ for some matrix $M$ such that scores $\widehat{\boldsymbol{s}}_t$ have the same distribution as some $\boldsymbol{c} + M\boldsymbol{z}$ with $\boldsymbol{z}$ following some spherical distribution. In particular, Assumption 3.8 and Property B.1 impose that $M$ is a $m \times m$ matrix and they entail that there exists $\sigma^2 > 0$ such that $\Sigma = \sigma^2 M M^\top = \sigma^2 \Gamma$.

Therefore, we actually have, when Algorithm 2 is run with any projection matrix $P$ onto $\mathrm{Im}(H)$,

$$\mathbb{E}\left[\sum_{i=1}^m w_i\, \ell\big(\widetilde{C}_i(\boldsymbol{x}_{T+1})\big)^2\right] = \left(\sum_{i=1}^m w_i\, \Gamma_{i,i}'\right) \mathbb{E}\big[L_\alpha^2\big] = \mathrm{Tr}\big(\mathrm{diag}(\boldsymbol{w})\, P\Sigma P^\top\big)\, \frac{\mathbb{E}\big[L_\alpha^2\big]}{\sigma^2}.$$

Lemma B.5 shows that $P_{\Sigma^{-1}}$ is a projection matrix onto $\mathrm{Im}(H)$. Lemma 3.9 below shows that for all projections $P$ onto $\mathrm{Im}(H)$ and all positive vectors $\boldsymbol{w} \in \mathbb{R}^m$,

$$\mathrm{Tr}\big(\mathrm{diag}(\boldsymbol{w})\, P_{\Sigma^{-1}} \Sigma P_{\Sigma^{-1}}^\top\big) \leqslant \mathrm{Tr}\big(\mathrm{diag}(\boldsymbol{w})\, P\Sigma P^\top\big).$$

Collecting all elements, whether $\mathbb{E}[L_\alpha^2] = +\infty$ or $\mathbb{E}[L_\alpha^2] \in [0, +\infty)$, we proved so far that when Algorithm 2 is run with any projection matrix $P$ onto $\mathrm{Im}(H)$ to output prediction intervals $\widetilde{C}_i$,

$$\forall \boldsymbol{w} \in (0, +\infty)^m, \qquad \mathbb{E}\left[\sum_{i=1}^m w_i\, \ell\big(\widetilde{C}_i^\star(\boldsymbol{x}_{T+1})\big)^2\right] \leqslant \mathbb{E}\left[\sum_{i=1}^m w_i\, \ell\big(\widetilde{C}_i(\boldsymbol{x}_{T+1})\big)^2\right]. \tag{14}$$

We obtain the claimed component-wise inequalities by taking $w_i = 1$ for one component $i$ and letting $w_j \to 0$ for $j \neq i$. $\quad\square$

We now move on to the proof of Corollary 3.11.

**Corollary 3.11.** *In the setting and under the assumptions of Theorem 3.10, we also have*

$$\forall i \in [m], \quad \mathbb{E}\Big[\ell\big(\widetilde{C}_i^\star(\boldsymbol{x}_{T+1})\big)^2\Big] \leqslant \mathbb{E}\Big[\ell\big(\widehat{C}_i(\boldsymbol{x}_{T+1})\big)^2\Big],$$

*where the prediction rectangle $\widehat{C}_1(\boldsymbol{x}_{T+1}) \times \ldots \times \widehat{C}_m(\boldsymbol{x}_{T+1})$ is output by the plain multivariate version of SCP (Algorithm 1).*

*Proof.* The result follows from Theorems 3.7 and 3.10 (which both hold under the stronger set of assumptions of Theorem 3.10). More precisely, for each $\boldsymbol{w} \in (0, +\infty)^m$, denote by $\widetilde{C}_i^{\boldsymbol{w}}$ the prediction intervals output by Algorithm 2 run with $P = P_{\boldsymbol{w}}$. Theorem 3.7 ensures that

$$\forall \boldsymbol{w} \in (0, +\infty)^m, \qquad \mathbb{E}\left[\sum_{i=1}^m w_i\, \ell\big(\widetilde{C}_i^{\boldsymbol{w}}(\boldsymbol{x}_{T+1})\big)^2\right] \leqslant \mathbb{E}\left[\sum_{i=1}^m w_i\, \ell\big(\widehat{C}_i(\boldsymbol{x}_{T+1})\big)^2\right].$$

Equality (14) in the proof of Theorem 3.10 states that

$$\forall \boldsymbol{w} \in (0, +\infty)^m, \qquad \mathbb{E}\left[\sum_{i=1}^m w_i\, \ell\big(\widetilde{C}_i^\star(\boldsymbol{x}_{T+1})\big)^2\right] \leqslant \mathbb{E}\left[\sum_{i=1}^m w_i\, \ell\big(\widetilde{C}_i^{\boldsymbol{w}}(\boldsymbol{x}_{T+1})\big)^2\right].$$

Combining these two inequalities, we have

$$\forall \boldsymbol{w} \in (0, +\infty)^m, \qquad \mathbb{E}\left[\sum_{i=1}^m w_i\, \ell\big(\widetilde{C}_i^\star(\boldsymbol{x}_{T+1})\big)^2\right] \leqslant \mathbb{E}\left[\sum_{i=1}^m w_i\, \ell\big(\widehat{C}_i(\boldsymbol{x}_{T+1})\big)^2\right],$$

and we conclude the proof with the same limit arguments as after (14) in the proof of Theorem 3.10. $\quad\square$

The following lemma is a deep and central result in the theory of forecast reconciliation. First stated for $W = \mathrm{Id}_m$ by Wickramasuriya et al. (2019), this result has since been extended to symmetric positive semi-definite matrices $W$ in Panagiotelis et al. (2021) and Ando & Narita (2024). We provide a short and elementary proof, which may actually be seen as a simplification of the proof by Ando & Narita (2024), an article entirely devoted to proving Lemma 3.9. The latter article sees the minimization problem at hand as a constrained minimization problem (given how projections onto $\mathrm{Im}(H)$ may be written), thus introduced a Lagrangian and discussed Karush-Kuhn-Tucker conditions to solve it.

**Lemma 3.9** (Minimum-Trace projection). *Let $W$ and $\Sigma$ be two symmetric $m \times m$ matrices, where $W$ positive semi-definite and $\Sigma$ is positive definite. Then, for all projection matrices $P$ onto $\mathrm{Im}(H)$,*

$$\mathrm{Tr}\big(W P_{\Sigma^{-1}} \Sigma P_{\Sigma^{-1}}^\top\big) \leqslant \mathrm{Tr}\big(W P \Sigma P^\top\big).$$

*Proof.* We first show that projection matrices $P$ onto $\mathrm{Im}(H)$ are exactly the matrices of the form $HG$, where $G$ is a $n \times m$ matrix such that $GH = \mathrm{Id}_n$. Indeed, such a matrix $HG$ satisfies $HG\,HG = HG$ and $HG\,H = H$, which characterizes projections onto $\mathrm{Im}(H)$. Conversely, fix a projection $P$ onto $\mathrm{Im}(H)$ and a basis $\boldsymbol{u}_1, \ldots, \boldsymbol{u}_m$ of $\mathbb{R}^m$: each $P\boldsymbol{u}_i$ belongs to $\mathrm{Im}(H)$, thus is of the form $H\boldsymbol{g}_i$ for some $\boldsymbol{g}_i \in \mathbb{R}^n$. Denote by $G$ the $n \times m$ matrix with columns given by $\boldsymbol{g}_1, \ldots, \boldsymbol{g}_m$. By linearity of $P$ and given that $\boldsymbol{u}_1, \ldots, \boldsymbol{u}_m$ is a basis, we have $P = HG$. We denote by $\boldsymbol{h}_1, \ldots, \boldsymbol{h}_n$ the columns of the $m \times n$ structural matrix $H$. Since $P$ is a projection onto $\mathrm{Im}(H)$, we have in particular $P\boldsymbol{h}_i = \boldsymbol{h}_i$ for all $i \in [n]$, or put differently, $PH = H$. Substituting $P = HG$ and multiplying both sides by $H^\top$, we proved so far that $H^\top HGH = H^\top H$, where (see Lemma B.4), the matrix $H^\top H$ is invertible. All in all, we thus proved $GH = \mathrm{Id}_n$.

Given the characterization above, the projection matrices $P$ onto $\mathrm{Im}(H)$ are also exactly the matrices of the form

$$P = P_{\Sigma^{-1}} + HA = H\left(\left(H^\top \Sigma^{-1} H\right)^{-1} H^\top \Sigma^{-1} + A\right), \qquad \text{for } n \times m \text{ matrices } A \text{ such that} \qquad AH = [0]_n\,,$$

where $[0]_n$ denotes the $n \times n$ null matrix. Keeping in mind that $\Sigma$ and $\Sigma^{-1}$ are symmetric, this decomposition entails that

$$
\begin{aligned}
WP\Sigma P^\top = \quad & W\left(H\left(H^\top \Sigma^{-1} H\right)^{-1} H^\top \Sigma^{-1}\right) \Sigma \left(\Sigma^{-1} H \left(H^\top H\right)^{-1} H^\top\right) \\
& + W\left(HA\right) \Sigma \left(\Sigma^{-1} H \left(H^\top \Sigma^{-1} H\right)^{-1} H^\top\right) \\
& + W\left(H\left(H^\top \Sigma^{-1} H\right)^{-1} H^\top \Sigma^{-1}\right) \Sigma \left(A^\top H^\top\right) \\
& + W\left(HA\right) \Sigma \left(HA\right)^\top.
\end{aligned}
$$

The second term in the decomposition simplifies into

$$W\left(HA\right)\Sigma\left(\Sigma^{-1} H \left(H^\top \Sigma^{-1} H\right)^{-1} H^\top\right) = WH\overbrace{AH}^{=[0]_n}\left(H^\top \Sigma^{-1} H\right)^{-1} H^\top = [0]_m\,.$$

Similarly, the third term is also null, due to the term $H^\top \Sigma^{-1} \Sigma A^\top = (AH)^\top$. The proof is concluded by noting that for all matrices $A$, the trace of the fourth term in the decomposition of $WP\Sigma P^\top$ is non-negative. Indeed, given that $W$ and $\Sigma$ are positive semi-definite, we may write them as $W = MM^\top$ and $\Sigma = NN^\top$ for $m \times m$ matrices $M$ and $N$. Then, together with elementary properties of the trace,

$$
\begin{aligned}
\mathrm{Tr}\left(W\left(HA\right)\Sigma\left(HA\right)^\top\right) = \mathrm{Tr}\left(MM^\top (HA)NN^\top (HA)^\top\right) &= \mathrm{Tr}\left(M^\top (HA)NN^\top (HA)^\top M\right) \\
&= \mathrm{Tr}\left(\left(M^\top (HA)N\right)\left(M^\top (HA)N\right)^\top\right) \geqslant 0\,,
\end{aligned}
$$

given that the trace of a symmetric positive semi-definite matrix is non-negative. $\qquad\square$

# D. Forecast reconciliation: review and connections made

For a complete review on the forecast reconciliation literature, we refer the reader to Athanasopoulos et al. (2024) and only provide a brief overview below.

At the origins, forecasts in the hierarchical setting were conducted using a single-level approach (most notably, in the bottom-up or top-down fashion), i.e., by choosing a level of the hierarchy (typically, either the bottom level or the top level) to generate forecasts, and then, by propagating these forecasts (typically in a linear fashion). A notable pitfall of the single-level approaches is that potentially valuable information from all other levels are ignored. To overcome this issue, the concept of forecast reconciliation was introduced by Athanasopoulos et al. (2009) and Hyndman et al. (2011): the idea is to combine forecasts from different levels of aggregation through linear combinations. Recently, developments were made in reconciliation through projections (Wickramasuriya et al., 2019, Panagiotelis et al., 2021), which we review and detail in the next section.

Probabilistic hierarchical forecasting and reconciliation is an emerging field. Notable works include the one by Wickramasuriya (2024), which studied probabilistic forecast reconciliation for Gaussian distributions, while Panagiotelis et al. (2023) provided reconciled forecasts based on the minimization of a probabilistic score through gradient descent. However, we did not leverage results from this literature for our own probabilistic approach.

## D.1. Review of forecast reconciliation through projections

We summarize and review the approach followed by Hyndman et al. (2011), Wickramasuriya et al. (2019), and Panagiotelis et al. (2021).

The setting is the one described in Section 2, with stochastic observations following some hierarchical structure $\boldsymbol{y} = H\boldsymbol{y}_{1:n}$; features are possibly available. Initial point forecasts $\widehat{\boldsymbol{y}}$ are provided by some regression method $\mathcal{A}$; these forecasts are possibly incoherent, i.e., do not belong to $\mathrm{Im}(H)$. The goal of forecast reconciliation is to leverage the hierarchical structure to improve the point forecasts.

A typical assumption made in this literature is that the point forecasts $\widehat{\boldsymbol{y}}$ are unbiased, or, put differently, that the forecasting errors $\widehat{\boldsymbol{s}} = \boldsymbol{y} - \widehat{\boldsymbol{y}}$ are centered. A natural performance criterion then is the mean-square error [MSE]: letting $\|\cdot\|_2$ denote the Euclidean norm and $\Sigma$ the covariance matrix of $\widehat{\boldsymbol{s}} = \boldsymbol{y} - \widehat{\boldsymbol{y}}$,

$$\text{MSE}(\widehat{\boldsymbol{y}}, \boldsymbol{y}) \stackrel{\text{def}}{=} \mathbb{E}\big[\|\widehat{\boldsymbol{s}}\|_2^2\big] = \mathbb{E}\big[\widehat{\boldsymbol{s}}^\top \widehat{\boldsymbol{s}}\big] = \mathbb{E}\Big[\text{Tr}\big(\widehat{\boldsymbol{s}}\,\widehat{\boldsymbol{s}}^\top\big)\Big] = \text{Tr}\Big(\mathbb{E}\big[\widehat{\boldsymbol{s}}\,\widehat{\boldsymbol{s}}^\top\big]\Big) \stackrel{\text{def}}{=} \text{Tr}(\Sigma)\,.$$

The equalities above may be generalized to $W$–norms (as defined in Lemma B.5), where $W$ is a symmetric definite positive matrix:

$$\text{MSE}(\widehat{\boldsymbol{y}}, \boldsymbol{y}, W) \stackrel{\text{def}}{=} \mathbb{E}\big[\|\widehat{\boldsymbol{s}}\|_W^2\big] = \mathbb{E}\big[\widehat{\boldsymbol{s}}^\top W \widehat{\boldsymbol{s}}\big] = \mathbb{E}\Big[\text{Tr}\big(W \widehat{\boldsymbol{s}}\widehat{\boldsymbol{s}}^\top\big)\Big] = \text{Tr}(W\Sigma)\,.$$

Natural improvements of the unbiased point forecasts are exactly given by projections thereof onto $\text{Im}(H)$, as justified below in Lemma D.1. Let $P$ be a projection onto $\text{Im}(H)$ and denote $\widetilde{\boldsymbol{y}} = P\widehat{\boldsymbol{y}}$. By linearity of a projection, the point forecasts $\widetilde{\boldsymbol{y}}$ are also unbiased. Since observations are coherent, we have

$$\boldsymbol{y} - \widetilde{\boldsymbol{y}} \stackrel{\text{def}}{=} \boldsymbol{y} - P\widehat{\boldsymbol{y}} = P\big(\boldsymbol{y} - \widehat{\boldsymbol{y}}\big) = P\widehat{\boldsymbol{s}} \stackrel{\text{def}}{=} \widetilde{\boldsymbol{s}}\,.$$

The mean-squared error of $\widetilde{\boldsymbol{y}}$ in $W$–norm thus equals

$$\text{MSE}(\widetilde{\boldsymbol{y}}, \boldsymbol{y}, W) = \mathbb{E}\Big[\text{Tr}\big(W \widetilde{\boldsymbol{s}}\,\widetilde{\boldsymbol{s}}^\top\big)\Big] = \text{Tr}\Big(W\,P\,\mathbb{E}\big[\widehat{\boldsymbol{s}}\,\widehat{\boldsymbol{s}}^\top\big]P^\top\Big) = \text{Tr}\big(W\,P\Sigma P^\top\big)\,.$$

Actually, the formula above holds more generally for all matrices $P$ such that $PH = H$.

Optimal unbiased point forecasts in the sense of the mean-square error thus exactly correspond to minimizing $\text{Tr}\big(W\,P\Sigma P^\top\big)$, a problem that we discuss below. Before we do so, we justify why (only) projections onto $\text{Im}(H)$ are considered.

**Why (only) projections onto $\text{Im}(H)$ are considered.** This follows from the lemma below, given that the literature of forecast reconciliation considers, implicitly or explicitly, two restrictions: that forecasts should be unbiased; that improved forecasts should be obtained by linear combinations of the original forecasts (and be coherent, of course).

**Lemma D.1** (Hyndman et al., 2011)**.** *Assume that the point forecasts $\widehat{\boldsymbol{y}}$ are unbiased. Let $M$ be a $m \times m$ matrix taking values in the coherent subspace $\text{Im}(H)$. Then the linear combinations $\widetilde{\boldsymbol{y}} = M\widehat{\boldsymbol{y}}$ are unbiased if and only if $M$ is a projection onto $\text{Im}(H)$.*

*Proof.* Being unbiased means the following in Hyndman et al. (2011): we denote by $\boldsymbol{m} = H\boldsymbol{\beta}$ the expectation of $\boldsymbol{y}$, i.e., $\mathbb{E}[\boldsymbol{y}] = \boldsymbol{m} = H\boldsymbol{\beta}$, and assume that the model is rich enough so that all values of $\boldsymbol{\beta} \in \mathbb{R}^n$, i.e., all values of $\boldsymbol{m} \in \text{Im}(H)$, may be obtained when the specifications of the model vary.

That $\widetilde{\boldsymbol{y}} = M\widehat{\boldsymbol{y}}$ is unbiased thus corresponds to the equalities

$$\forall \boldsymbol{\beta} \in \mathbb{R}^n, \quad MH\boldsymbol{\beta} = H\boldsymbol{\beta}\,, \qquad \text{i.e.,} \qquad MH = H.$$

Now, the proof of Lemma 3.9 in Appendix C shows that since $M$ takes values in $\text{Im}(H)$, it is of the form $M = HG$ for some $n \times m$ matrix $G$. The equality $MH = H$ may be rewritten as $HGH = H$. Again as in the proof of Lemma 3.9, by multiplying both sides of this equality by $(H^\top H)^{-1}H^\top$, we obtain $GH = \text{Id}_n$, which yields $M^2 = HGHG = HG = M$. Thus, $M$ is indeed a projection onto $\text{Im}(H)$. $\square$

**Trace optimization, part 1: known covariance matrix.** As explained above, original unbiased forecasts $\widehat{\boldsymbol{y}}$ and their (still unbiased, linear) transformations $\widetilde{\boldsymbol{y}} = P\widehat{\boldsymbol{y}}$, where $P$ is a projection onto $\text{Im}(H)$ may be compared through their mean-squared errors in $W$–norm:

$$\text{MSE}(\widehat{\boldsymbol{y}}, \boldsymbol{y}, W) = \text{Tr}(W\Sigma) \qquad \text{vs.} \qquad \text{MSE}(\widetilde{\boldsymbol{y}}, \boldsymbol{y}, W) = \text{Tr}\big(W\,P\Sigma P^\top\big)\,.$$

This consideration leads to the central result in forecast reconciliation: the optimality of the so-called Minimum Trace reconciliation method from Wickramasuriya et al. (2019), formally re-stated below as Lemma 3.9. This method consists of projecting according to $P_{\Sigma^{-1}} \stackrel{\text{def}}{=} H\big(H^\top \Sigma^{-1} H\big)^{-1} H^\top \Sigma^{-1}$, where we recall that $\Sigma$ is the (unknown) covariance matrix of the forecast errors $\widehat{\boldsymbol{s}}$. Of course, this theoretically optimal method must be turned into a practical method, e.g., by replacing $\Sigma$ in the formula above by some empirical estimate.

Lemma 3.9 was originally stated by Wickramasuriya et al. (2019) in the case $W = \mathrm{Id}_m$, and later extended to symmetric positive semi-definite matrices $W$ by Panagiotelis et al. (2021) and Ando & Narita (2024). As discussed in Appendix C, we provide a more elementary proof.

**Lemma 3.9** (Minimum-Trace projection). *Let $W$ and $\Sigma$ be two symmetric $m \times m$ matrices, where $W$ positive semi-definite and $\Sigma$ is positive definite. Then, for all projection matrices $P$ onto $\mathrm{Im}(H)$,*

$$\mathrm{Tr}\big(W P_{\Sigma^{-1}} \Sigma P_{\Sigma^{-1}}^{\top}\big) \leqslant \mathrm{Tr}\big(W P \Sigma P^{\top}\big).$$

**Trace optimization, part 2: a more practical approach.**  The drawback with the approach above is that it relies on the knowledge of the covariance matrix $\Sigma$, but its advantage is that it holds for all weight matrices $W$. We now show how to exchange the roles of $W$ and $\Sigma$, and get a trace-reduction result for a given weight matrix $W$ but for all possible covariance matrices $\Gamma$, i.e., symmetric positive semi-definite matrices.

This result is inspired from Panagiotelis et al. (2021), who recommend to use the orthogonal projection in $W$–norm, whose closed-form expression (see Lemma B.5) reads $P_W \stackrel{\text{def}}{=} H\big(H^{\top}WH\big)^{-1}H^{\top}W$. A Pythagorean theorem ensures that, for all point forecasts $\widehat{y}$ and (coherent) observations $y$,

$$\big\| y - P_W \widehat{y} \big\|_W^2 = \big\| P_W(y - \widehat{y}) \big\|_W^2 \leqslant \big\| (y - \widehat{y}) \big\|_W^2 \quad \text{a.s.,}$$

thus, by taking expectations,

$$\mathrm{Tr}\big(W\, P_W \Sigma P_W^{\top}\big) = \mathrm{MSE}\big(P_W \widehat{y}, y, W\big) \leqslant \mathrm{MSE}\big(\widehat{y}, y, W\big) = \mathrm{Tr}(W\Sigma).$$

The equality above holds no matter the specific value of the covariance matrix $\Sigma$, which corresponds to the following trace-reduction inequality, stated as the second part of Lemma B.5: for all symmetric positive semi-definite matrices $\Gamma$,

$$0 \leqslant \mathrm{Tr}\big(W P_W \Gamma P_W^{\top}\big) \leqslant \mathrm{Tr}\big(W\Gamma\big). \tag{15}$$

The inequality above (i.e., Lemma B.5) is a result of our own though it was inspired by both Lemma 3.9 and the approach by Panagiotelis et al. (2021) relying on $P_W$–projections.

### D.2. How we leveraged and transferred these results (and why it was not immediate)

**On the unnecessity of unbiasedness.**  As we made clear several times in Section D.1, a key assumption in the literature of forecast reconciliation is that point forecasts are unbiased, or put differently, that the forecasting errors $\widehat{s}$ are centered.

This is in sharp contrast with the non-conformity scores $\widehat{s}$ considered in this article, which we do not want (nor need) to assume are centered. None of Assumptions 3.1–3.6–3.8 are about this. We rather assume that these scores follows a so-called elliptic distribution, with possibly a non-null expectation. Elliptic distributions were considered, not in the literature of reconciliation of point forecasts but of probabilistic forecasts, see Panagiotelis et al. (2023). Now, the proof of Theorem 3.7 in Appendix B reveals that our construction of prediction rectangles is such that the length of the $i$–th defining interval is given by

$$\ell\big(\widehat{C}_i(\boldsymbol{x}_{T+1})\big) = \widehat{s}_{\big(\lceil (T_{\text{calib}}+1)(1-\alpha/2)\rceil\big),i} - \widehat{s}_{\big(\lfloor (T_{\text{calib}}+1)\alpha/2\rfloor\big),i}.$$

Non-null expectations of the underlying elliptic distribution cancel out in the above equation, hence the unnecessity of an assumption of unbiasedness. The cancellation is only possible because we considered signed non-conformity scores (which is unusual in the literature of conformal prediction).

**On the component-wise objective** $(\star\star)$.  As we detail in Appendix E (see the comments before Theorem E.3), the theory provided in this article is only worth being detailed because we do not target joint-coverage guarantees but component-wise coverage guarantees. We had to find out a component-wise efficiency objective that we could handle. With the literature of forecast reconciliation in mind, we somehow had to build an intuition a such an efficiency criterion.

The proof of Theorem 3.7 in Appendix B explains how we could relate our (component-wise) small-length objective $(\star\star)$, namely,

$$\text{minimizing} \quad \mathbb{E}\left[\sum_{i=1}^m w_i\, \ell\big(C_i(\boldsymbol{x}_{T+1})\big)^2\right], \tag{16}$$

to problems of the form

$$\text{minimizing} \quad \text{Tr}\big(\text{diag}(\boldsymbol{w})\, P\Gamma P\big), \tag{17}$$

for some symmetric positive semi-definite matrix $\Gamma$, so as to leverage inequality (15), which is of our own. The proof of Theorem 3.10 reveals that when non-conformity scores have a bounded second-order moment, the matrix $\Gamma$ is proportional to their covariance matrix $\Sigma$, which opened the avenue of the Minimum Trace approaches of Lemma 3.9.

**Summary of the challenges overcome.**   In a nutshell, the main challenge overcome was to relate the two minimization problems (16) and (17), and in the first place, state suitably the efficiency criterion (16). The main tools used were to resort to signed non-conformity scores, which are not necessarily unbiased, and to exploit properties of elliptic distributions, in terms of stability of the shapes of these distributions under certain affine transformations.

## E. Multi-target split conformal prediction: Reminder and extension

Conformal prediction is a framework initially introduced to quantity the uncertainty around univariate targets thanks to univariate non-conformity scores. In this section, we focus on extensions of conformal prediction to multivariate targets as introduced by Johnstone & Cox (2021) and Messoudi et al. (2022). The key idea of these methods is to consider $A$–norms of non-conformity scores, where $A$ a data-based definite positive matrix designed to capture the potential multivariate dependencies of the targets. (The choice of $A$ is the key for efficient prediction regions in practice; see the statement of Lemma B.5 for a reminder of the definition of $A$–norms.) In a nutshell, and as formally detailed below, the consideration of data-based norms effectively matches vector predictions to scalar non-conformity scores.

### E.1. Overview of existing results

We first state the objectives and then review the methodology followed. We consider multivariate data (features $\boldsymbol{x}_t \in \mathbb{R}^d$ and observations $\boldsymbol{y}_t \in \mathbb{R}^m$) but with no specific hierarchical structure—unlike in Section E.2 below.

We now denote the prediction regions by $E(\boldsymbol{x})$ because they will typically be given by ellipsoids. (Messoudi et al., 2022 empirically illustrated that ellipsoidal predictive regions are more efficient than hyper-rectangular ones in terms of volumes.)

**Objectives.**   The cited references are interested in joint coverage guarantees and replace the component-wise coverage objective ($\star$) by the design of prediction regions $E : \boldsymbol{x} \mapsto E(\boldsymbol{x}) \subseteq \mathbb{R}^m$ such that

$$\mathbb{P}\big(\boldsymbol{y}_{T+1} \in E(\boldsymbol{x}_{T+1})\big) \approx 1 - \alpha\,, \tag{$\diamond$}$$

where the probability $\mathbb{P}$ is with respect to both $(\boldsymbol{x}_{T+1}, \boldsymbol{y}_{T+1})$ and $(\boldsymbol{x}_t, \boldsymbol{y}_t)_{1 \leqslant t \leqslant T}$.

The secondary objective of ensuring that the prediction sets output are efficient, i.e., as small as possible. We could state it through volumes, as given by the Lebesgue measure $\mathfrak{L}$ over $\mathbb{R}^m$, and replace objective ($\star\star$) by

$$\text{minimizing} \quad \mathbb{E}\Big[\mathfrak{L}\big(E(\boldsymbol{x}_{T+1})\big)\Big]\,, \tag{$\diamond\diamond$}$$

where again, the expectation is with respect to both $(\boldsymbol{x}_{T+1}, \boldsymbol{y}_{T+1})$ and $(\boldsymbol{x}_t, \boldsymbol{y}_t)_{1 \leqslant t \leqslant T}$. In Theorem E.3, we are actually able to prove an even stronger result of uniform domination: a prediction region $E$ is uniformly more efficient than a prediction region $E'$ if $E(\boldsymbol{x}) \subseteq E'(\boldsymbol{x})$ for all $\boldsymbol{x} \in \mathbb{R}^d$.

**Methodology and algorithm.**   We actually present a generalization of the methodology and algorithm considered by Johnstone & Cox (2021), in the spirit of Algorithm 3. The proof of the joint coverage result below, Theorem E.2, only relies on Assumption E.1, which holds as long as the matrix $A$ is chosen based on data of $\mathcal{D}_{\text{train}}$ and $\mathcal{D}_{est}$ only (just as was the case in Section 3.4).

The exact procedure is stated in Algorithm 4. A regressor $\widehat{\boldsymbol{\mu}}$ is built based on the data of the train set $\mathcal{D}_{\text{train}}$ and on a regression algorithm $\mathcal{A}$. Then, based on data $\mathcal{D}_{\text{train}}$ and $\mathcal{D}_{\text{estim}}$, but preferably only on $\widehat{\boldsymbol{\mu}}$ and on data from $\mathcal{D}_{\text{estim}}$, some symmetric definite positive matrix $A$ is computed in some black-box fashion; this is why Algorithm 4 takes a procedure $\mathcal{E}$ as input. This matrix $A$ is used to define a weighted norm (see Lemma B.5):

$$\boldsymbol{u} \in \mathbb{R}^m \longmapsto \|\boldsymbol{u}\|_A \stackrel{\text{def}}{=} \sqrt{\boldsymbol{u}^\top A \boldsymbol{u}}\,.$$

---

**Algorithm 4** Multivariate SCP through data-based norms

---

**Parameters:** confidence level $1 - \alpha$; regression algorithm $\mathcal{A}$; partition of $[T]$ into subsets $\mathcal{D}_{\text{train}}$, $\mathcal{D}_{\text{estim}}$ and $\mathcal{D}_{\text{calib}}$ of respective cardinalities $T_{\text{train}}$, $T_{\text{estim}}$ and $T_{\text{calib}}$; estimation procedure $\mathcal{E}$ of the matrix used to define the norm

1: Build the regressor $\widehat{\boldsymbol{\mu}}(\,\cdot\,) = \mathcal{A}\big((\boldsymbol{x}_t, \boldsymbol{y}_t)_{t \in \mathcal{D}_{\text{train}}}\big)$
2: Compute a symmetric definite positive matrix $A = \mathcal{E}\big((\boldsymbol{x}_t, \boldsymbol{y}_t)_{t \in \mathcal{D}_{\text{train}} \cup \mathcal{D}_{\text{estim}}}\big)$
3: **for** $t \in \mathcal{D}_{\text{calib}}$ **let** $\widehat{\boldsymbol{y}}_t = \widehat{\boldsymbol{\mu}}(\boldsymbol{x}_t)$ and $\check{s}_t = \|\boldsymbol{y}_t - \widehat{\boldsymbol{y}}_t\|_A$
4: Order the $(\check{s}_t)_{t \in \mathcal{D}_{\text{calib}}}$ into $\check{s}_{(1)} \leqslant \ldots \leqslant \check{s}_{(T_{\text{calib}})}$ and define $\check{s}_{(0)} = 0$ and $\check{s}_{(T_{\text{calib}}+1)} = +\infty$
5: Let $\check{q}_{1-\alpha} = \check{s}_{\big(\lceil (T_{\text{calib}}+1)(1-\alpha) \rceil\big)}$
6: Set $\check{E}(\,\cdot\,) = \Big\{ \boldsymbol{y} \in \mathbb{R}^m : \big\|\boldsymbol{y} - \widehat{\boldsymbol{\mu}}(\,\cdot\,)\big\|_A \leqslant \check{q}_{1-\alpha} \Big\}$
7: **return** $\check{E}(\boldsymbol{x}_{T+1})$

---

The multivariate prediction errors $\boldsymbol{y}_t - \widehat{\boldsymbol{\mu}}(\boldsymbol{x}_t)$ for $t \in \mathcal{D}_{\text{calib}}$ are then transformed into the univariate non-conformity scores through $A$–norms:

$$\check{s}_t = \|\boldsymbol{y}_t - \widehat{\boldsymbol{y}}_t\|_A \,.$$

Some threshold $\check{q}_{1-\alpha}$ is determined based on the empirical quantiles of the series of these scores, and finally, the following prediction regions are output, which are ellipsoids:

$$\check{E}(\,\cdot\,) = \Big\{ \boldsymbol{y} \in \mathbb{R}^m : \big\|\boldsymbol{y} - \widehat{\boldsymbol{\mu}}(\,\cdot\,)\big\|_A \leqslant \check{q}_{1-\alpha} \Big\} \,.$$

For instance, Johnstone & Cox (2021) use the so-called Mahalanobis distance, which corresponds to taking $A$ as the inverse of the covariance matrix of the forecasting errors (in the same spirit as in Algorithm 3).

**Coverage guarantees.** They rely on an i.i.d. assumption on the non-conformity scores (just as for Theorem 3.2 and as, more generally, in the literature of conformal prediction).

**Assumption E.1.** The non-conformity scores $\check{s}_t = \|\boldsymbol{y}_t - \widehat{\boldsymbol{\mu}}(\boldsymbol{x}_t)\|_A$ are i.i.d. for $t \in \mathcal{D}_{\text{calib}} \cup \{T+1\}$. This is in particular the case when data $(\boldsymbol{x}_t, \boldsymbol{y}_t)_{1 \leqslant t \leqslant T+1}$ is i.i.d.

**Theorem E.2.** *Fix* $\alpha \in (0, 1)$*. Algorithm 4, used with any regression algorithm $\mathcal{A}$ and any estimation procedure $\mathcal{E}$, ensures that whenever Assumption E.1 (i.i.d. scores) holds,*

$$\mathbb{P}\big(\boldsymbol{y}_{T+1} \in \check{E}(\boldsymbol{x}_{T+1})\big) \geqslant 1 - \alpha \,.$$

*In addition, if the non-conformity scores $\big(\check{s}_t\big)_{t \in \mathcal{D}_{\text{calib}} \cup \{T+1\}}$ are almost surely distinct, then*

$$\forall i \in [m], \mathbb{P}\big(\boldsymbol{y}_{T+1} \in \check{E}(\boldsymbol{x}_{T+1})\big) \leqslant 1 - \alpha + \frac{1}{T_{\text{calib}} + 1} \,.$$

*Proof.* We first note that by definition,

$$\Big\{ \boldsymbol{y}_{T+1} \in \check{E}(\boldsymbol{x}_{T+1}) \Big\} = \Big\{ \check{s}_{T+1} \leqslant \check{s}_{\big(\lceil (T_{\text{calib}}+1)(1-\alpha) \rceil\big)} \Big\} \,.$$

The rest of the proof consists of the same classical arguments that were already detailed in the proof of Theorem 3.2 in Appendix A. □

### E.2. Extension to hierarchical data

We now assume that the observations $\boldsymbol{y}_t$ follow a hierarchical structure, as in (1). We adapt Algorithm 4 into Algorithm 5 in the same way we obtained Algorithm 2 from Algorithm 1, by merely adding a projection step right before computing the non-conformity scores on $\mathcal{D}_{\text{calib}}$. We resort to the orthogonal projection matrix onto $\text{Im}(H)$ in $\|\cdot\|_A$–norm, whose closed-form expression reads (see Lemma B.5)

$$P_A = H(H^\top A H)^{-1} H^\top A \,.$$

---

**Algorithm 5** Hierarchical SCP through data-based norms

---

**Parameters:** confidence level $1 - \alpha$; regression algorithm $\mathcal{A}$; partition of $[T]$ into subsets $\mathcal{D}_{\text{train}}$, $\mathcal{D}_{\text{estim}}$ and $\mathcal{D}_{\text{calib}}$ of respective cardinalities $T_{\text{train}}$, $T_{\text{estim}}$ and $T_{\text{calib}}$; estimation procedure $\mathcal{E}$ of the matrix used to define the norm

1: Build the regressor $\widehat{\boldsymbol{\mu}}(\,\cdot\,) = \mathcal{A}\big((\boldsymbol{x}_t, \boldsymbol{y}_t)_{t \in \mathcal{D}_{\text{train}}}\big)$
2: Compute a symmetric definite positive matrix $A = \mathcal{E}\big((\boldsymbol{x}_t, \boldsymbol{y}_t)_{t \in \mathcal{D}_{\text{train}} \cup \mathcal{D}_{\text{estim}}}\big)$
3: Let $P_A = H(H^\top A H)^{-1} H^\top A$ and consider $\mathring{\boldsymbol{\mu}}(\,\cdot\,) = P_A \widehat{\boldsymbol{\mu}}(\,\cdot\,)$
4: **for** $t \in \mathcal{D}_{\text{calib}}$ **let** $\mathring{\boldsymbol{y}}_t = \mathring{\boldsymbol{\mu}}(\boldsymbol{x}_t)$ and $\mathring{s}_t = \|\boldsymbol{y}_t - \mathring{\boldsymbol{y}}_t\|_A$
5: Order the $(\mathring{s}_t)_{t \in \mathcal{D}_{\text{calib}}}$ into $\mathring{s}_{(1)} \leqslant \ldots \leqslant \mathring{s}_{(T_{\text{calib}})}$ and define $\mathring{s}_{(0)} = 0$ and $\mathring{s}_{(T_{\text{calib}}+1)} = +\infty$
6: Let $\mathring{q}_{1-\alpha} = \mathring{s}_{\big(\lceil (T_{\text{calib}}+1)(1-\alpha)\rceil\big)}$
7: Set $\mathring{E}(\,\cdot\,) = \Big\{ \boldsymbol{y} \in \mathbb{R}^m : \big\|\boldsymbol{y} - \widehat{\boldsymbol{\mu}}(\,\cdot\,)\big\|_A \leqslant \mathring{q}_{1-\alpha} \Big\}$
8: **return** $\mathring{E}(\boldsymbol{x}_{T+1})$

---

Instead of considering the regressor $\widehat{\boldsymbol{\mu}}$, which may yield point estimates not abiding by the hierarchical constraints, we resort to $\mathring{\boldsymbol{\mu}} = P_A \widehat{\boldsymbol{\mu}}$. The rest of the procedure is similar to the previous section.

We summarize the adaptation in Algorithm 5. The statement only differs from Algorithm 4 by the addition of the blue line. We denote by $\mathring{s}$ and $\mathring{E}$ the scores and prediction ellipsoids obtained after the projection step, to distinguish them from the ones obtained by Algorithm 4 without the projection step.

We may prove the following result, which shows that the objectives ($\diamondsuit$) and ($\diamondsuit\diamondsuit$) are met. Its proof is straightforward. Put differently, there would have been no challenge in providing a theory of efficient conformal prediction for hierarchical data under a joint-coverage objective ($\diamondsuit$). This was not the case at all for component-wise coverage objectives, as the tools of forecast reconciliation (like the projections step by $P_A$) are not component-wise tools. The proof of Theorem E.3 actually emphasizes the complexity of results such as Theorems 3.10–3.7 and Corollary 3.11.

**Theorem E.3.** *Fix $\alpha \in (0, 1)$. Algorithm 5, used with any regression algorithm $\mathcal{A}$ and any estimation procedure $\mathcal{E}$, guarantees the same coverage results as in Theorem E.2 whenever Assumption E.1 (i.i.d. scores) holds.*

*In addition, the prediction ellipsoids $\mathring{E}$ output by Algorithm 5 are uniformly more efficient than the prediction ellipsoids $\check{E}$ output by Algorithm 4:*

$$\mathring{E}(\boldsymbol{x}_{T+1}) \subseteq \check{E}(\boldsymbol{x}_{T+1}) \quad a.s.$$

*Proof.* The proof of the coverage guarantees is similar to the one in Theorem E.2. For the uniform efficiency part, we first note by a Pythagorean theorem, and since observations are coherent, $\mathring{s}_t \leqslant \check{s}_t$ for all $t \in \mathcal{D}_{\text{calib}}$. Thus, in particular

$$\mathring{s}_{\big(\lceil (T_{\text{calib}}+1)(1-\alpha)\rceil\big)} \leqslant \check{s}_{\big(\lceil (T_{\text{calib}}+1)(1-\alpha)\rceil\big)},$$

hence, the stated inclusion, by construction of the predictive regions. $\qquad\square$

# F. Full details for the simulations: settings, methodology, results

In this appendix, we provide the full details on the specifications and of the results of the numerical experiments summarized in the main body of the article.

## F.1. Artificially generated data

The objective of the experiments on synthetic data is to replicate the behavior of real data while controlling the number of observations available and the difficulty of the forecasting task.

### F.1.1. DATA GENERATION

**Data generation: initial draw of the parameters.** The structural matrix $H$ considered in this example is the right one in Figure 1; we copy it in Figure 3 for the convenience of the reader. Therefore, there are $m = 8$ nodes in the hierarchy, with $n = 5$ nodes at the most disaggregated levels. For each given run, we first pick at random a function $f : \mathbb{R}^3 \to \mathbb{R}^5$ and a covariance matrix $A^\top A$. We do so as explained later in this description.

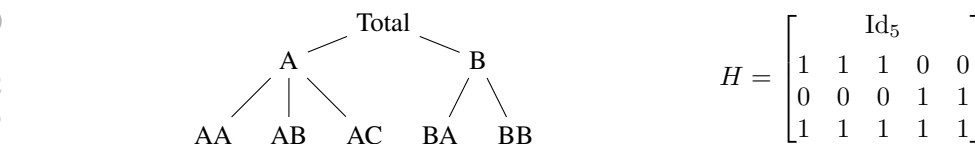

*Figure 3.* The structural matrix $H$ considered in the numerical experiments with artificially generated data.

**Data generation: draw of $T$–sample.** Then, given $H$, $f$, and $A$, we draw a $T$–sample $(\boldsymbol{x}_t, \boldsymbol{y}_t)_{1 \leqslant t \leqslant T}$ of data as follows. First, the features $\boldsymbol{x}_t \in \mathbb{R}^3$ are drawn i.i.d. according to a Gaussian distribution:

$$\boldsymbol{x}_t = \begin{bmatrix} x_{t,1} \\ x_{t,2} \\ x_{t,3} \end{bmatrix} \sim \mathcal{N}\left( \begin{bmatrix} 1 \\ 0 \\ -1 \end{bmatrix}, \begin{bmatrix} 2 & 0 & 0 \\ 0 & 2 & 0 \\ 0 & 0 & 1 \end{bmatrix} \right).$$

Next, the observations $\boldsymbol{y}_{t,1:5} \in \mathbb{R}^5$ at the most disaggregated level are generated i.i.d. according to the following additive model:

$$\boldsymbol{y}_{t,1:5} = f(\boldsymbol{x}_t) + \boldsymbol{\varepsilon}_t, \qquad \text{where} \qquad \boldsymbol{\varepsilon}_t \sim \mathcal{N}\left( \begin{bmatrix} 10 \\ \vdots \\ 10 \end{bmatrix}, A^{\top}A \right). \tag{18}$$

The complete vectors of observations are finally given by $\boldsymbol{y}_t = H\boldsymbol{y}_{t,1:5}$.

**Data generation: initial draw of the parameters, continued.** The matrix $A$ is drawn component-wise, in an i.i.d. manner: the $A_{i,j}$, where $i, j \in [5]$, follow a standard Gaussian distribution $\mathcal{N}(0,1)$.

We draw $f = (f_1, \ldots, f_5)$ component-wise. To do so, we consider the following base functions $\mathbb{R}^3 \to \mathbb{R}$:

$$
\begin{aligned}
&g_1(\boldsymbol{x}_t) = x_{1,t}, && g_5(\boldsymbol{x}_t) = x_{2,t}, && g_9(\boldsymbol{x}_t) = x_{3,t}, \\
&g_2(\boldsymbol{x}_t) = x_{1,t}^2, && g_6(\boldsymbol{x}_t) = x_{2,t}^2, && g_{10}(\boldsymbol{x}_t) = x_{3,t}^2, \\
&g_3(\boldsymbol{x}_t) = \sin(x_{1,t}), && g_7(\boldsymbol{x}_t) = \cos(x_{2,t}), && g_{11}(\boldsymbol{x}_t) = \exp(x_{3,t}), \\
&g_4(\boldsymbol{x}_t) = \log\left(|x_{1,t}| + 1\right), && g_8(\boldsymbol{x}_t) = \sqrt{x_{2,t}}.
\end{aligned}
$$

We now explain how $f_i$ is drawn for each component $i \in [5]$. First, the number $k_i$ of effects to consider is drawn uniformly in the set $[11]$. Then, we sample with replacement $k_i$ base functions in the set $\{g_1, \ldots, g_{11}\}$; we denote them by $h_{i,1}, \ldots, h_{i,k_i}$. Finally, we add signs: we draw $k_i$ i.i.d. symmetric Rademacher random variables $r_{i,1}, \ldots, r_{i,k_i}$ (i.e., variables that take values $-1$ and $1$ with respective probabilities $1/2$). All in all, we let

$$f_i = \sum_{j=1}^{k_i} r_{i,j}\, h_{i,j}.$$

F.1.2. DATA SPLITTING

We take $T = 100\,000$ (to contrast with the experiments on real data). These $T$ observations are first randomly split in two subsets, containing $80\%$ and $20\%$ of the data.

The smaller subset is referred to as the test set and is denoted by $\mathcal{D}_{\text{test}}$. Its data points will play the role of the $(\boldsymbol{x}_{T+1}, \boldsymbol{y}_{T+1})$, as explained later in Appendix F.1.4.

The larger subset of $80\%$ of the data is split again in three sub-subsets, containing $40\%$ (train set $\mathcal{D}_{\text{train}}$), $20\%$ (estimation set $\mathcal{D}_{\text{estim}}$), and $20\%$ (calibration set $\mathcal{D}_{\text{calib}}$) of the total data. These data points are used to construct the prediction rectangles, which are all (in Algorithms 1–2–3) of the form

$$\boldsymbol{x} \longmapsto \prod_{i=1}^{8} \widetilde{C}_i(\boldsymbol{x}) = \prod_{i=1}^{8} \left[ \widetilde{\mu}_i(\boldsymbol{x}) + \widetilde{q}_{\alpha/2}^{(i)}, \, \widetilde{\mu}_i(\boldsymbol{x}) + \widetilde{q}_{1-\alpha/2}^{(i)} \right],$$

and only depend on the features $\boldsymbol{x}$ through the centers $\widetilde{\mu}_i(\boldsymbol{x})$. The algorithms that do not use an estimation set $\mathcal{D}_{\text{estim}}$, i.e., Algorithms 1–2, simply ignore data points in $\mathcal{D}_{\text{estim}}$.

F.1.3. TRAIN SET: REGRESSION ALGORITHM $\mathcal{A}$

The last piece to fully define the procedures implemented is to describe the regression algorithm $\mathcal{A}$ given as input to Algorithms 1–2–3. This algorithm will be given by a base forecasting method run independently at each node.

Before we describe this base forecasting method, we mention a constraint that we impose. It turns out that in the practice of hierarchical forecasting, explanatory variables are not necessarily all available at every level of granularity within the hierarchical structure. (For instance, some meteorological data may only be available at specific locations equipped with the necessary sensors and cannot be communicated in a timely, real-time, manner to other nodes.) This also actually makes the hierarchy more interesting from a forecasting viewpoint since the observations at some nodes are harder to predict than others.

To reproduce this specificity, for each of the nodes at the most disaggregated level, indexed by $i \in [5]$, we draw independently a Bernoulli variable $\rho_i$ with parameter 0.7: if $\rho_i = 1$, then the forecasting strategy may use the entire vectors $\boldsymbol{x}_t$; otherwise, the forecasting strategy only accesses to $\boldsymbol{x}'_t = (x_{t,1}, x_{t,2})^\top$.

It only remains to describe the forecasting strategy used independently at each node $i \in [8]$, based on features that lie in $\mathbb{R}^2$ or $\mathbb{R}^3$. Given the additive nature (18) of the data, a natural choice is to resort to the theory of estimation of generalized additive models, see a reminder at the end of this subsection.

For each $i \in [8]$, depending on $\rho_i$, the regression estimate $\widehat{\mu}_i$ produced for the $i$–th component of the $\boldsymbol{y}$ is of the form

$$\widehat{\mu}_i : \boldsymbol{x} \longmapsto \begin{cases} \widehat{\mu}_i^{(1)}(x_1) + \widehat{\mu}_i^{(2)}(x_2) + \widehat{\mu}_i^{(3)}(x_3), & \text{if } \rho_i = 1 \\ \widehat{\mu}_i^{(1)}(x_1) + \widehat{\mu}_i^{(2)}(x_2), & \text{otherwise.} \end{cases}$$

**Reminder on generalized additive models.** Generalized additive models (GAMs, Wood, 2017) are a popular modeling for electricity demand. They form a good compromise between forecast efficiency and interpretability. In that setting, univariate response variables $z_t$ based on features $\boldsymbol{x}_t \in \mathbb{R}^d$, where $t \in [T]$, are expressed as

$$z_t = \beta_0 + \sum_{j=1}^{d} m_j(x_{t,j}) + \varepsilon_t, \tag{19}$$

where the $m_j : \mathbb{R} \to \mathbb{R}$ do not depend on $t$ and are called the non-linear effects, and where the $\varepsilon_t$ are i.i.d. random noises. The non-linear effects $m_j$ are each possibly decomposed on a given spline basis $(B_{j,k})$, chosen by the forecasting agent:

$$m_j : x \in \mathbb{R} \longmapsto \sum_{k=1}^{K_j} \beta_{j,k} B_{j,k}(x),$$

where $K_j$ depends on the dimension of the spline basis. Estimating the model (19) then amounts to estimating the coefficients $\beta_{j,k}$.

At a high level, we may write that the estimation of these coefficients $\beta_{j,k}$ is performed via by penalized least-squares, where the penalty term therein involves the second derivatives of the functions $m_j$, forcing the effects to be smooth. We resorted to the R package mgcv of Wood (2023) in our simulations, with the basis by default: the thin plate spline basis, with a maximum number of degrees of freedom of 10.

F.1.4. TEST SET: EVALUATION OF THE PREDICTION SETS

The objectives (⋆) and (⋆⋆) are both in terms of coverage probability and expected length with respect to all data (the observations to be predicted as well as the data used to compute the prediction intervals). We consider, for the expected-length criterion (⋆⋆), weights given by the constant vector $\boldsymbol{w} = (1, \dots, 1)^\top$.

On the test set, we estimate the conditional coverage probabilities and expected lengths given the specifications of the experiment (i.e., $f$, $A$, and the $\rho_i$) and given data in the sets $\mathcal{D}_{\text{train}}$, $\mathcal{D}_{\text{estim}}$, and $\mathcal{D}_{\text{calib}}$: for each $i \in [8]$,

$$c_i \stackrel{\text{def}}{=} \frac{1}{T_{\text{test}}} \sum_{t \in \mathcal{D}_{\text{test}}} \mathbb{1}_{\left\{ y_{t,i} \in \widetilde{C}_i(\boldsymbol{x}_t) \right\}} \qquad \text{and} \qquad \ell_i \stackrel{\text{def}}{=} \ell\big(\widetilde{C}_i(\cdot)\big),$$

where we denoted by $T_{\text{test}} = 20\,000$ the cardinality of $\mathcal{D}_{\text{test}}$. That is, for the conditional coverage probability, we resort to Monte-Carlo-type estimates. For the lengths of the intervals $\widetilde{C}_i(\boldsymbol{x})$, we note that they do not depend on $\boldsymbol{x}$, so are constant on $\mathcal{D}_{\text{test}}$; we denote by $\ell\big(\widetilde{C}_i(\,\cdot\,)\big)$ their common value.

We actually run the entire procedure a large number of times to get unconditional probabilities and expectations, as described next.

### F.1.5. Monte-Carlo estimates based on large numbers of runs

We run $1\,000$ the entire procedure and get, for each run, estimates of the conditional coverage probabilities and expected lengths, which we denote by:

$$c_i^{(r)} \quad \text{and} \quad \ell_i^{(r)}, \qquad \text{where} \quad i \in [8] \ \text{and} \ r \in [1\,000].$$

We in turn get the following estimates for the unconditional coverage probabilities and expected squared lengths: for each $i \in [8]$,

$$\overline{c}_i \stackrel{\text{def}}{=} \frac{1}{1\,000} \sum_{r=1}^{1\,000} c_i^{(r)} \qquad \text{and} \qquad \overline{\ell}_i \stackrel{\text{def}}{=} \frac{1}{1\,000} \sum_{r=1}^{1\,000} \big(\ell_i^{(r)}\big)^2.$$

These empirical means estimate the underlying unconditional coverage probabilities and expected squared lengths up to 95%–confidence errors margins given by

$$\gamma_{c,i} \stackrel{\text{def}}{=} 1.96\, \frac{\text{std}\Big(c_i^{(1)}, \,\ldots,\, c_i^{(1\,000)}\Big)}{\sqrt{1\,000}} \qquad \text{and} \qquad \gamma_{\ell,i} \stackrel{\text{def}}{=} 1.96\, \frac{\text{std}\Big(\big(\ell_i^{(1)}\big)^2, \,\ldots,\, \big(\ell_i^{(1\,000)}\big)^2\Big)}{\sqrt{1\,000}},$$

where $\text{std}(x_1, \ldots, x_{1\,000})$ denotes the standard deviation of the data series given as argument:

$$\text{std}(x_1, \ldots, x_{1\,000}) = \sqrt{\frac{1}{1\,000} \sum_{r=1}^{1\,000} \big(x_r - \overline{x}_{1\,000}\big)^2}, \qquad \text{where} \quad \overline{x}_{1\,000} = \frac{1}{1\,000} \sum_{r=1}^{1\,000} x_r.$$

For scaling issues on the lengths, we rather report, in our experiments, when dealing with component-wise quantities, the following point estimates and associated confidence intervals on the underlying unconditional probabilities and expectations: for all $i \in [8]$,

$$\overline{c}_i \quad \text{and} \quad \sqrt{\overline{\ell}_i}, \qquad\qquad \big[\overline{c}_i \pm \gamma_{c,i}\big] \quad \text{and} \quad \Big[\sqrt{\overline{\ell}_i - \gamma_{\ell,i}}, \sqrt{\overline{\ell}_i + \gamma_{\ell,i}}\Big]. \tag{20}$$

### F.1.6. Component-wise results: coverage and length

The top graph of Figure 4 reports the indicators defined in (20), with standard errors in $x$–axis corresponding to the estimation of the component-wise coverage levels, and standard errors in $y$–axis, to the ones for the lengths. The comments on the outcomes are to be found in Section 4.

### F.1.7. Global results: total lengths

We also report results on the total lengths, i.e., for the quantities appearing in the efficiency objective ($\star\star$), where we recall that $\boldsymbol{w} = \mathbf{1}$.

In the same spirit as in Section F.1.5, we consider

$$L_{\bullet}^{(r)} = \sum_{i=1}^{8} \big(\ell_i^{(r)}\big)^2, \quad \text{where} \ r \in [1\,000], \qquad \overline{L}_{\bullet} \stackrel{\text{def}}{=} \frac{1}{1\,000} \sum_{r=1}^{1\,000} L_{\bullet}^{(r)} = \sum_{i=1}^{8} \overline{\ell}_i.$$

This empirical mean estimates the underlying expected sum of the squared lengths up to 95%–confidence errors margins given by

$$\gamma_{L,\bullet} \stackrel{\text{def}}{=} 1.96\, \frac{\text{std}\Big(L_{\bullet}^{(1)}, \,\ldots,\, L_{\bullet}^{(1\,000)}\Big)}{\sqrt{1\,000}},$$

For the same scaling issues as in section F.1.5, we rather report in our experiments the following point estimates and associated confidence intervals:

$$\sqrt{\overline{L}_\bullet}, \qquad \left[\sqrt{\overline{L}_\bullet - \gamma_{L,\bullet}}, \sqrt{\overline{L}_\bullet + \gamma_{L,\bullet}}\right]. \tag{21}$$

The results are represented in the bottom graph of Figure 4, with comments to be found in Section 4.

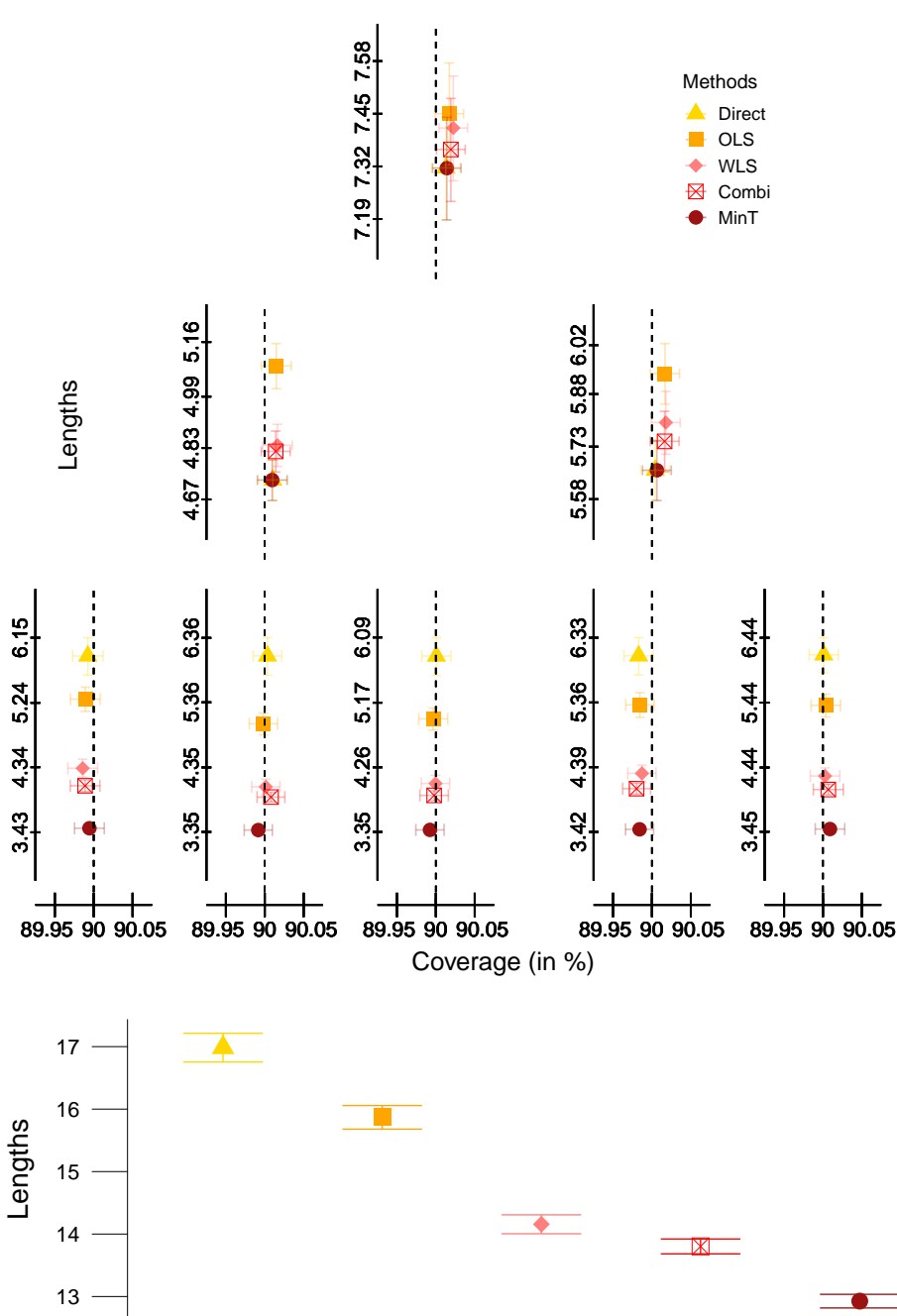

*Figure 4.* Artificially generated data: component-wise coverage levels and prediction-interval lengths (*top figure*) and total lengths (*bottom figure*). This figure merely performs a zoom on the left graphs of Figure 2. The standard errors reported are based on the formulas (20) and (21).

### F.2. Palo Alto daily charging Energy

The dataset we consider is presented in greater detail by Amara-Ouali et al. (2021), an article referencing several data sets for charging sessions of electric vehicles. We refer to this data set as the Palo Alto dataset, as it is related to charging stations located in the city of Palo Alto, CA. It contains a substantial number of charging sessions and an interesting hierarchical structure, with 47 charging points that are divided into a dozen of stations. However, due to some real-world considerations, data is not available for all these charging points in the 2015–2019 period considered. We only consider 2 charging stations (called "Riconada Library" and "Hamilton", featuring 3 and 2 charging points, respectively) for which data is available on the entire period. The study stops in 2019 to avoid the temporary shift in demand caused by Covid19 in 2020 and the subsequent years.

We are interested in daily energy demand at each charging point, each charging station, and at the global level of the considered hierarchy. The total number of observations for each node is $T = 1\,780$.

#### F.2.1. METHODOLOGY

The methodology followed for this data set essentially mimics the one for artificial data, as presented in Section F.1. We thus present only the main adaptations made.

**Runs and data splitting.** We will perform 360 runs of a given experimental procedure, described next. We split into train, estimation, calibration, and test sets as in Section F.1.2, with same 40%–20%–20%–20% proportions.

**Regression algorithm $\mathcal{A}$.** As in Section F.1.3 (and taking inspiration from Amara-Ouali et al., 2022), we resort to modeling and forecasting through GAMs. More specifically, for each node $i \in [8]$ and day $t$, we consider the following auto-regressive specification of GAM for the energy demand $y_{t,i}$:

$$y_{t,i} = \beta^{(0)} + \sum_{j=1}^{7} \beta_j^{(1)} \mathbb{1}_{\text{DayType}_t = j} + \sum_{j=1}^{7} m_1(y_{t-1}) \mathbb{1}_{\text{DayType}_t = j} \tag{22}$$

$$+ m_2(y_{t-1,i}) + m_3(y_{t-7,i}) + m_4(t) + m_5(\text{ToY}_t) + \varepsilon_t, \tag{23}$$

where $\text{DayType}_t \in \{1, \ldots, 7\}$ is a categorical variable indicating the day of the week, $\text{ToY}_t$ is the "time of year", i.e., the position of the day in the year (whose value grows linearly from 0 on the 1st of January to 1 on the 31st of December). The model (22) incorporates a trend term $m_4(t)$, which may be estimated because we pick a random subset of the entire data set for the train set; this trend term is useful to take into account changes in the infrastructures and shifts in user behaviors.

We again resorted the R package `mgcv` of Wood (2023) to forecast this model and get $\widehat{\mu}$. The parameters were the default thin plate spline basis and a maximum number of degrees of freedom of 10 for the estimation of the coefficients for $m_1$, $m_2$ $m_3$, and 15 for $m_4$. We fitted $m_5$ with cyclic splines and a maximum number of degrees of freedom of 30.

**Validity check of Assumption 3.6 on elliptic distribution.** Based on the regression function $\widehat{\mu}$ output, we computed the signed non-conformity scores on the estimation and calibration sets. We then fitted a Student distribution (which is a particular case of an elliptic distribution). Figures 5 and 6 illustrate the goodness of fit between the empirical distributions of scores and the Student distributions with parameters estimated on these scores, through, respectively, densities and Q-Q–plots. We were interested in two nodes $i$: the total node (the total demand for the 5 charging points) and the node of the Riconada Library (the sum of the 3 charging points located there).

The fit to a Student distribution looks reasonable in both cases, with actually an excellent fit for the total node and a relatively small mode of the distribution located in the right tail of the distribution scores being not captured well in the case of the Riconada Library node.

**Estimation set: details on the "Oracle MinT" strategy.** For this data set, we also report the performance of the "Oracle MinT" strategy, i.e., Algorithm 2 used with $P_{\Sigma^{-1}}$. The question is how to determine $\Sigma$.

The "MinT" strategy, i.e., Algorithm 3 with $\mathcal{P}_{\text{MinT}}$, truly estimates $\Sigma$, on the estimation set:

$$\widehat{\Sigma} = \frac{1}{T_{\text{estim}}} \sum_{t \in \mathcal{D}_{\text{estim}}} \left(\widehat{s}_t - \overline{s}\right)\left(\widehat{s}_t - \overline{s}\right)^\top, \qquad \text{where} \qquad \overline{s} = \frac{1}{T_{\text{estim}}} \sum_{t \in \mathcal{D}_{\text{estim}}} \widehat{s}_t.$$

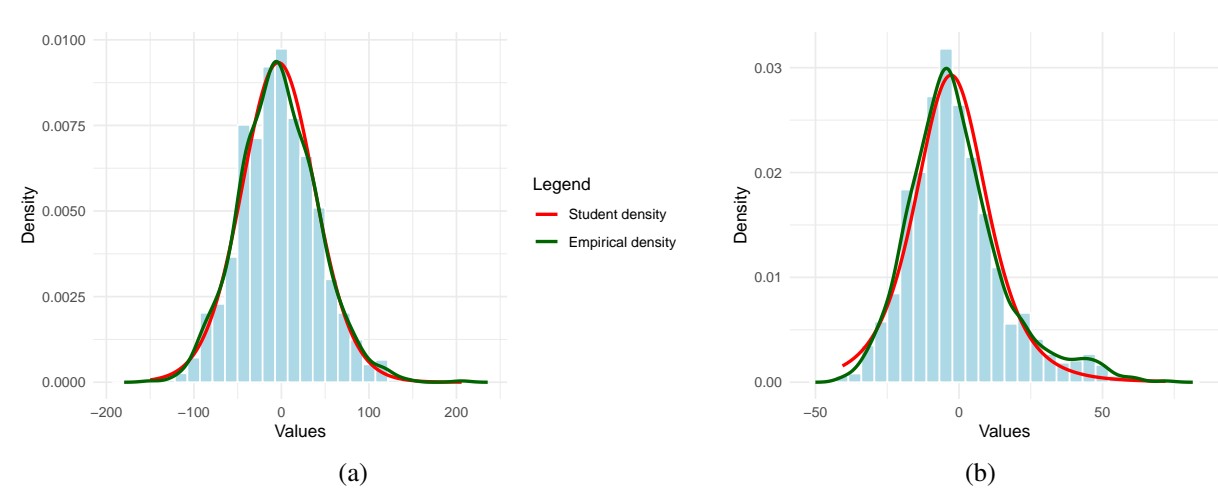

*Figure 5.* Student density estimator compared to the empirical density of the scores for the total node (a) and the Rinconada Library station (b).

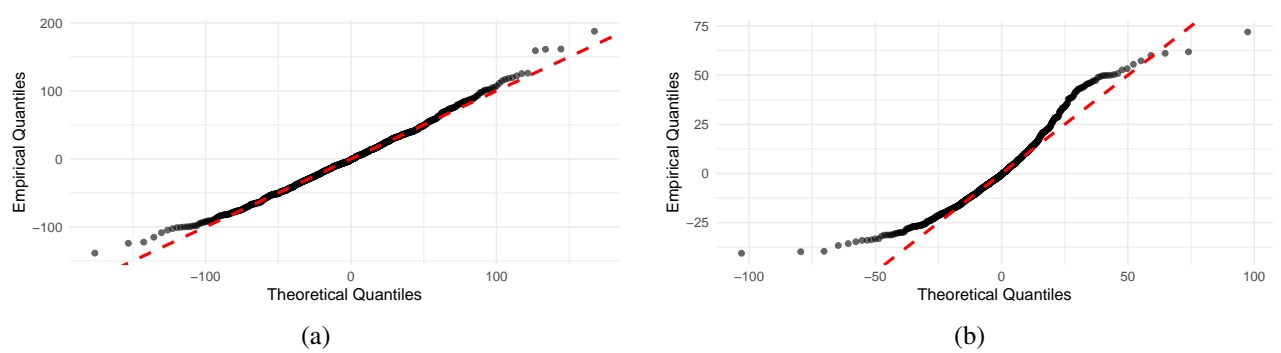

*Figure 6.* Q-Q–plots of the scores for the total node (a) and the Rinconada Library station (b).

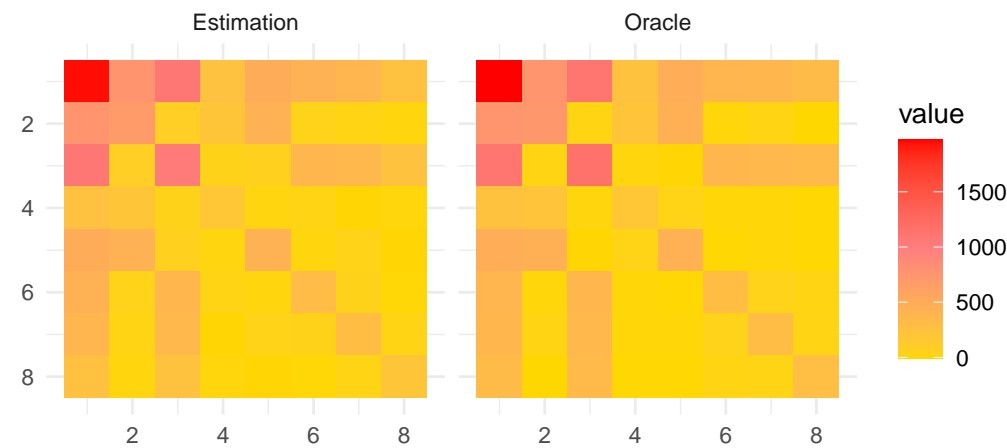

*Figure 7.* The estimates $\widehat{\Sigma}$ and $\widehat{\Sigma}^{\star}$ of the covariance matrix of the non-conformity scores for one given run of the experiment.

In the case of the Palo Alto data set, the results obtained by this strategy are rather poor, which may be attributed to having few data points only to perform the estimation: the ones of the estimation set $\mathcal{D}_{\text{estim}}$.

To determine $\Sigma$ for the "Oracle MinT" strategy, we actually also estimate it, but by cheating: we produce an estimate thereof using $\mathcal{D}_{\text{calib}}$ and $\mathcal{D}_{\text{test}}$, the two subsets on which the non-conformity scores will be calculated. More precisely, we produce the estimate

$$\widehat{\Sigma}^{\star} = \frac{1}{T_{\text{calib}} + T_{\text{test}}} \sum_{t \in \mathcal{D}_{\text{calib}} \cup \mathcal{D}_{\text{test}}} \left(\widehat{s}_t - \overline{s}^{\star}\right)\left(\widehat{s}_t - \overline{s}^{\star}\right)^{\top}, \qquad \text{where} \qquad \overline{s}^{\star} = \frac{1}{T_{\text{calib}} + T_{\text{test}}} \sum_{t \in \mathcal{D}_{\text{calib}} \cup \mathcal{D}_{\text{test}}} \widehat{s}_t \,,$$

and run Algorithm 2 with $P_{(\widehat{\Sigma}^{\star})^{-1}}$.

Figure 7 displays the two estimates $\widehat{\Sigma}$ and $\widehat{\Sigma}^{\star}$ just for one given run picked at random out of the 360 runs. The differences between the two estimates look mild, yet, the differences in performance are substantial (see Figure 8). We repeated the comparison several times and always obtained quite similar estimates.

### F.2.2. OUTCOMES

We use the exact same metrics as in Sections F.1.5 and F.1.7, with the mere replacement of the number 1 000 of runs considered therein by the number 360 of runs considered now, and report the corresponding results in Figure 8. This figure merely performs a zoom on the right graphs of Figure 2. The comments on the outcomes are located in Section 4.

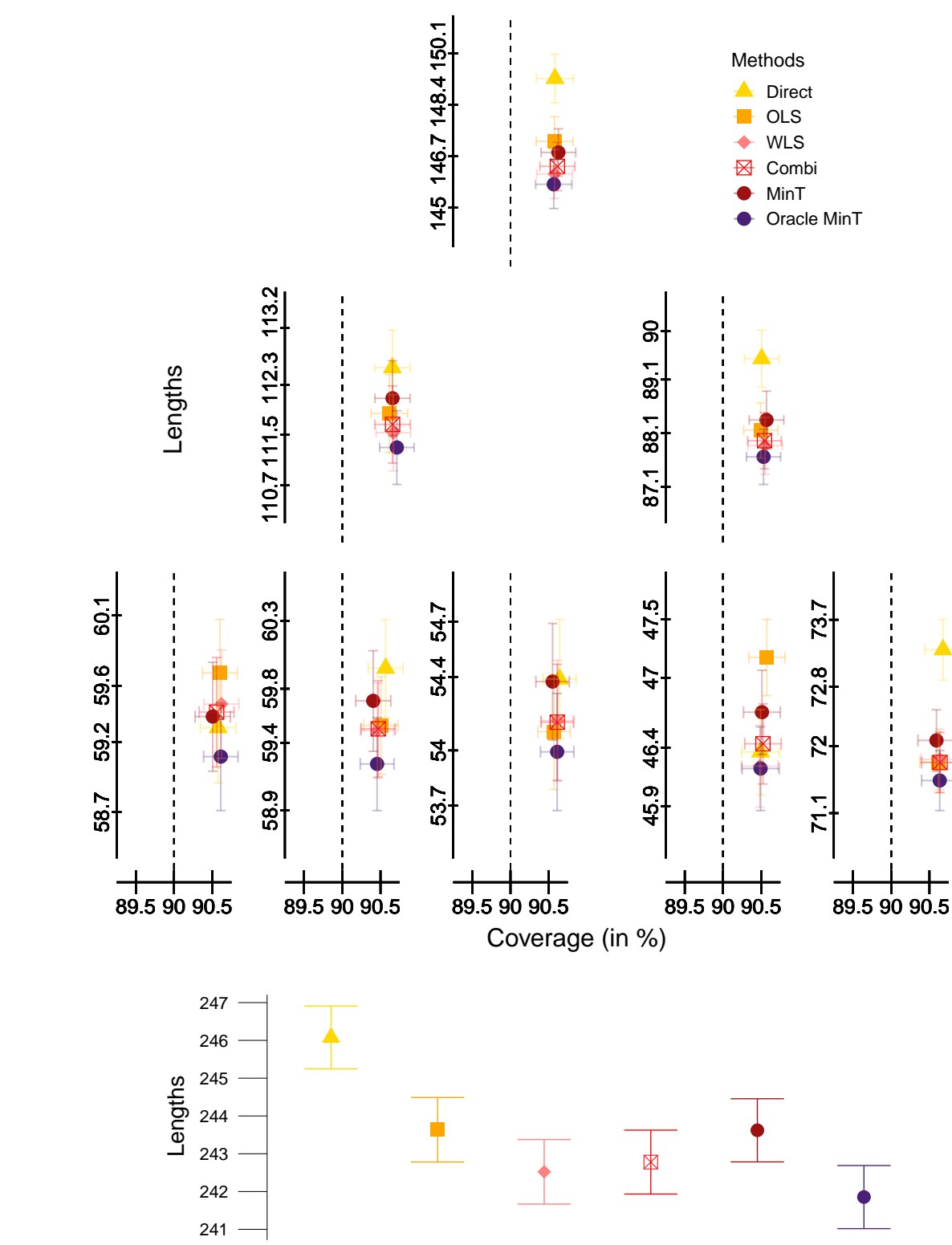

*Figure 8.* Palo Alta data of daily energy demand for charging electric vehicles: component-wise coverage levels and prediction-interval lengths (*top figure*) and total lengths (*bottom figure*). This figure merely performs a zoom on the right graphs of Figure 2. The standard errors reported are based on the formulas (20) and (21), with 1 000 replaced by 360.

