# OpenReview forum: "Conformal Prediction for Hierarchical Data"
_ICML.cc/2025/Conference — Submitted to ICML 2025_

### Official Review · Reviewer_KKXz · 2025-03-03

**Overall Recommendation:** 1

**Summary:**

This paper studies the problem of conformal prediction for hierarchically-structured multivariate regression datasets. Hierarchical coherence of the different outputs is encoded via a projection matrix. A novel split conformal prediction algorithm is introduced with two objectives in mind: marginal coverage for each of the output dimensions, and small intervals for each of the output dimensions.

The authors present extensive theoretical results and limited experimental results for their work. The main theoretical result is Theorem 3.7, which shows that the hierarchical conformal prediction algorithm presented by the authors leads to a smaller expected set size compared to the plain multivariate version. In the experiments a synthetic dataset and one real-world dataset are considered.

**Claims And Evidence:**

I have concerns about the theoretical and experimental claims. See below for more details.

**Essential References Not Discussed:**

No missing references.

**Experimental Designs Or Analyses:**

Apart from the remark on the non i.i.d. nature of the real-world dataset, I don't have other remarks about the experiments.

**Methods And Evaluation Criteria:**

I find the experimental results quite weak. Only one real-world dataset is considered, and this dataset is not convincing for me. Energy-demand forecasting is a time series problem, so the nonconformity scores will not be i.i.d. This is problematic, since the authors are not capable of showing that their method is useful for a practical problem. Overall, the motivation for extending conformal prediction to the hierarchical multivariate case is quite weak.

Are there actually real-world static regression problems with i.i.d. data and multivariate hierarchically-structured outputs? The minimum I expect is that at least one problem of that kind is studied in the experiments.

**Other Comments Or Suggestions:**

In Section 2.2 I did not understand the motivation for using signed nonconformity scores. It is unclear to me what advantage these have over absolute residuals. Please motivate better.

In the introduction the notion "data series" is somewhat misleading. Initially I thought that the paper was about hierarchically-structured time series (as in the energy-demand dataset).

**Other Strengths And Weaknesses:**

Strengths:
- The theoretical analysis looks strong
- The paper is well written (from a math perspective)
- The authors understand the topic very well

Weaknesses:
- Very specific problem setting
- Experiments and assumptions for theory not convincing
- The paper is quite technical

**Questions For Authors:**

The authors are welcome to give feedback to my comments.

**Relation To Broader Scientific Literature:**

Looks ok.

**Theoretical Claims:**

I did not check the proofs carefully. (I have to review 6 papers for ICML, so I don't have time to read appendices or proofs)

In Theorem 3.7 it is not clear to me whether the theoretical improvement over the plain multivariate conformal prediction algorithm comes from Assumption 3.6 or from imposing the hierarchical structure. My gut feeling is that it comes from Assumption 3.6. It is quite obvious that, with stronger assumptions than i.i.d. data or exchangeability on the nonconformity scores, stronger theoretical guarantees in terms of coverage or interval length can be obtained. Can theoretical improvement of the presented method be proven without introducing additional assumptions? In fact the elliptical distribution assumption of the nonconformity scores is a very strong assumption that is somewhat against the spirit of the conformal prediction literature (i.e. distribution-free guarantees).

---

> ### Author Rebuttal · Authors · 2025-03-31
>
> We thank the reviewer for the detailed comments; as listed in the strengths, our main objective was indeed to develop a theory (of conformal prediction for hierarchical data). This endeavour turns out to be much different from the theory of conformal prediction for multivariate data as we detail in Issue 3 below.
>
> **Issue 1. Very specific problem setting**
>
> Real-world static regression problems that fit the setting exist --- for instance, survey data with hierarchically structured answers (e.g., household budget surveys, where expenditures are decomposed accross various categories, as https://ec.europa.eu/eurostat/web/microdata/household-budget-survey), and more generally, any regression task for which the answers are divided into subcategories. We did not consider such real-world data sets because of the wish of testing robustness detailed below.
>
> &nbsp;
>
> **Issue 2. Experiments**
>
> The study with synthetic data is meant to exactly illustrate our theoretical claims, while the study based on the real data set shows that the results are robust in practice and should hold beyond the i.i.d case (and they also show that the assumption of elliptical distribution assumption is reasonable, see Figure 5 page 28). Admittedly, we however did not underline enough that the real case was for the sake of testing robustness, and will correct that. Our approach with this article is basically to derive theoretical results in an ideal setting (i.i.d and elliptically-distributed forecast errors) so as to establish the theoretical foundations for future research.
>
> &nbsp;
>
> **Issue 3.  Assumptions for theory**
>
> **3.1** Actually, the most important source for the improvement is the hierarchical structure. This is best illustrated in Appendix E, where the objective is about joint coverage and where the improvement in efficiency is achieved with no assumption on the distribution of scores (see Theorem E.3).
>
> **3.2** Now, the main challenge in our setting is that we do not just want some joint coverage, but target a more ambitious goal of coverage for each variable i of the hierarchy, referred to as individual coverages in the sequel. (This wish of individual coverages makes sense for hierarchical data but would not make sense for plain multivariate data.) For individual-coverages results, we do not need any assumption beyond the typical i.i.d. assumption.
>
> **3.3** The efficiency results for individual-coverages guarantees, however, do require both, and equally importantly, the hierarchical structure (as in the case of joint coverage) and the distributional assumption of elliptical distribution (as in the literature of forecast reconciliation, where it is standard). Fully distribution-free results would of course have been prefered, but first, elliptical distributions form a vast set of distributions, second, appear naturally as distributions for residuals, and third, results of efficiency in conformal prediction are most of the time only empirical results while theoretically-grounded such results are also only achieved under additional assumptions. One of the (very) few examples of such theoretical results consists of the article by Le Bars and Humbert (2025, On volume minimization in conformal regression) and requires additional assumptions on the forecast model. (We prefer assumptions on the distribution of scores.) Another reference is Dhillon et al. (AIStats 2024, On the expected size of conformal prediction sets): they pave a way for obtaining theoretical results on the efficiency in terms of some complex multiplicative factor suffering from complex interdependancies between the base forecasting method, the choice of the score, and the distribution of data --- controlling this factor would require complex additionnal assumptions.
>
> **3.4** Most of the articles on conformal prediction for multivariate data (including Johnstone and Cox, 2021, and Messoudi et al., 2022, which we both cite) derived empirical efficiency results based on the intuition that elliptical predictive regions will fit well the underlying distribution. To us, this intuition hints at an implicit assumption of elliptical distribution of the scores (which we made explicit).
>
> **3.5** All in all, we commit to incorporate the comments above in a revision, and to better insist on the differences in nature between the aims of joint coverage and individual coverages.
>
> &nbsp;
>
> **Why signed non-conformity scores:** From a practical viewpoint, absolute residuals produce prediction sets centered on the forecasts, which are poor when forecasts are biased. From a technical viewpoint, the projection of signed non-conformity scores corresponds to the non-conformity score of the projected forecasts, which is important in our proofs (e.g., in the first lines of the proof of Theorem 3.2, in Appendix A). Absolute values being non-linear, we do not have that the projection of absolute non-conformity scores correspond to the absolute values of the projected forecast errors.

---

> > ### Comment · Reviewer_KKXz · 2025-04-02
> >
> > I thank the authors for their detailed response. It made a few things more clear to me.

---

### Official Review · Reviewer_mvP5 · 2025-03-08

**Overall Recommendation:** 2

**Summary:**

This paper addresses an important question in distribution-free inference--constructing prediction intervals for the multivariate response. Motivated by forecast reconciliation, this paper proposes to utilize the hierarchical information among multivariate responses to enhance the accuracy of prediction sets.

**Claims And Evidence:**

Claims are supported by both theoretical and numerical results.

**Essential References Not Discussed:**

There are other related papers on conformal prediction for multivariate responses:
1. Xu, Chen, Hanyang Jiang, and Yao Xie. "Conformal prediction for multi-dimensional time series by ellipsoidal sets." arXiv preprint arXiv:2403.03850 (2024).
2. Dheur, Victor, Matteo Fontana, Yorick Estievenart, Naomi Desobry, and Souhaib Ben Taieb. "Multi-Output Conformal Regression: A Unified Comparative Study with New Conformity Scores." arXiv preprint arXiv:2501.10533 (2025).
3. Henderson, Iain, Adrien Mazoyer, and Fabrice Gamboa. "An adaptive covariance based score for conformal inference in multivariate regression." (2024).

**Experimental Designs Or Analyses:**

Yes.

**Methods And Evaluation Criteria:**

Yes.

**Other Comments Or Suggestions:**

N/A

**Other Strengths And Weaknesses:**

Strength
1. This paper is well-organized, technically solid, and studies an important question.
2. The connection built between conformal prediction and forecast reconciliation is novel.

Weakness
1. overall, the contribution of this paper is limited to the literature. Essentially, the projection of scores can be roughly related to the de-correlation of multivariate scores, which is actually already explored in the literature of conformal prediction (please see aforementioned papers).
2. The assumption that scores are from an elliptical distribution is limited, which may hurt the distribution-free nature of conformal prediction.

**Questions For Authors:**

1. Section 2-Settings. Is $m \geq 3$ essential for the theoretical guarantee? More concretely, when $m=2$, can we still apply the current approach?
2. The final output is a hyper rectangular. I wonder if this could be optimal. Intuitively speaking, if score vector $s$ is a multivariate Gaussian random vector with mean vector $0$ and covariance matrix $\Sigma$, the projection we are looking for should transform the covariance to identity, in which sense, this "reconciliation" approach is the same with "de-correlation". In addition, scores based on Mahalanobis norms are explored in the aforementioned papers, and it would be beneficial tp compare the current approach with those scores empirically.

**Relation To Broader Scientific Literature:**

Constructing distribution-free prediction sets for multivariate random vectors is important in practice.

**Theoretical Claims:**

Yes.

---

> ### Author Rebuttal · Authors · 2025-03-31
>
> We thank the reviewer for the detailed reading, and also for considering the connections built between conformal prediction and forecast reconciliation as one of the strengths of this submission. We go over the two issues pointed out.
>
> &nbsp;
>
> **Issue 1 - Projection vs. de-correlation and answer to Question 2**
>
> **1.1** The articles cited are all about joint coverage, an objective that we acknowledge before Equation (2) and discuss in detail in Appendix E. We will discuss these articles in these places (and also in the introduction). However, we must underline that joint coverage is a much different objective than the individual coverages stated in Equation (*). It is easy to leverage the hierarchical structure for the objective of joint coverage (see Appendix E), but less so for individual coverages. To us, targeting joint coverage or individual coverages are two distinct problems, which call for different approaches, though possibly based on similar underlying intuitions.
>
> **1.2** Now, about de-correlation and the central part of Question 2: does the reviewer refer to using the oracle projection matrix stated in Eq. (6) of Section 3.3? If so, our answer is that the two approaches are conceptually very different or that at least, we do not see the connection. Indeed, when H is the identity matrix (which corresponds to the classic multivariate setting with no hierarchical constraints), then the projection (6) is the identity and the covariance matrix is not involved at all.
>
> **1.3** As discussed below in our answer to Question 1, forecast reconciliation can be interpreted as forecast combination under constraints. Thus, using the oracle projection matrix stated in Eq. (6) rather corresponds to projecting without changing too much the nodes where the forecast errors are low compared to the others, which is encoded by the covariance matrix (and not to a de-correlation).
>
> **1.4** To now answer the first part of Question 2, we target hyper-rectangles because we want individual coverage guarantees (for each element i of the hierarchy), and such individual guarantees make sense for hierarchical data (while they would not for truly multivariate data). We thus cannot wonder what the optimal shape of the prediction region is: it has to be some hyper-rectangle.
>
> **1.5** For the last part of Question 2, note that we consider scores based on the Mahalanobis norms in Appendix E, as they relate to joint coverage -- but did not consider them in our experiments, which focus on individual coverages.
>
> **1.6** All in all, we will incorporate the comments above and better insist on the differences in nature between the aims of joint coverage and individual coverages.
>
> &nbsp;
>
> **Issue 2 - Elliptical distribution for scores**
>
> **2.1** Yes, fully distribution-free results would of course have been prefered, but first, elliptical distributions form a vast set of distributions, second, appear naturally as distributions for residuals, and third, results of efficiency in conformal prediction are most of the time only empirical results while theoretically-grounded such results are also only achieved under additional assumptions. One of the (very) few examples of such theoretical results consists of the article by Le Bars and Humbert (arXiv 2025, On Volume Minimization in Conformal Regression) and requires additional assumptions on the forecast model. (We prefer assumptions on the distribution of scores.) Another reference is Dhillon et al. (AIStats 2024, On the expected size of conformal prediction sets): they pave a way for obtaining theoretical results on the efficiency in terms of some multiplicative factor but the latter suffers from complex interdependancies between the base forecasting method, the choice of the score, and the distribution of data --- handling this multiplicative factor would require complex additionnal assumptions to get any efficiency results (and it is not even clear which assumptions would work).
>
> **2.2** Most of the articles on conformal prediction for multivariate data (including Johnstone and Cox, 2021, and Messoudi et al., 2022, which we both cite, and the articles mentionned by the reviewer) derived empirical efficiency results based on the intuition that elliptical predictive regions will fit well the underlying distribution. To us, this intuition hints at an implicit assumption of elliptical distribution of the scores (which we made explicit).
>
> &nbsp;
>
> **Question 1:** Yes, the theoretical results hold for m=2 and are actually interesting: in this case, the observations are equal for both nodes but the base forecasts can be different, and thus, our approach corresponds to finding the best combination of two forecasts, (where 'best' is relative in our setting to the interval lengths). The links between forecast combination and forecast reconciliation were studied, as far as point forecasts are concerned (not conformal prediction) by Hollyman et al. (EJOR 2021, Understanding forecast reconciliation).

---

> > ### Comment · Reviewer_mvP5 · 2025-04-06
> >
> > Thank you for the clarification! I have another question regarding the coverage guarantee: as the individual guarantee for multivariate response is related to individual p-values in multiple testing problems, if we are interested in a set of coordinates in Y and would like to combine those individual prediction sets, would it be loose in terms of coverage guarantee. Overall, the authors’ response is very helpful and I will adjust my score based on later discussion with other reviewers.

---

> > > ### Author Response · Authors · 2025-04-07
> > >
> > > Indeed, Bonferroni-correction approaches (or copula-based approaches) can be used to obtain simultaneous coverage guarantees for several (and possibly all) coordinates.  However, the resulting prediction regions, for similar joint coverage levels, are likely to be less efficient as their shapes is more constrained (e.g., they would be rectangles in case Bonferroni corrections are used).

---

### Official Review · Reviewer_4SfW · 2025-03-14

**Overall Recommendation:** 4

**Summary:**

The authors extend the conformal prediction framework to hierarchical data, defined as multivariate data where some variates are linear combinations of covariates. The authors establish tighter efficiency bounds on the size of the conformal intervals, and also establish new bounds on a component-wise coverage objective. They build upon split conformal prediction, and make a simple modification by construing a projection matrix $P$ to take the original SCP prediction $\hat{u}$ to a full hierarchical prediction $\tilde{u}$. This is inspired by existing literature in forecast reconciliation.

**Claims And Evidence:**

Yes. All claims made are well-supported by clear proofs.

**Essential References Not Discussed:**

There is no related-works section, although the introduction gives a comprehensive exposition.

**Experimental Designs Or Analyses:**

The experimental design was done rigorously, with 1000 runs over artificial data as well as 360 runs over real data. However, the experiments are lacking in breadth, as both settings only consider the somewhat trivial 5-2-1 hierarchy. It would be more interesting to conduct a scaling experiment to see how the proposed algorithms scale in both time and sample efficiency with respect to greater complexity of hierarchical levels.

**Methods And Evaluation Criteria:**

Yes.

**Other Comments Or Suggestions:**

N/A

**Other Strengths And Weaknesses:**

N/A

**Questions For Authors:**

N/A

**Relation To Broader Scientific Literature:**

This paper helps unite the previous somewhat disparate areas of multivariate conformal prediction and forecast reconciliation. In addition, they introduce for the first time the component-wise coverage objective, which will be a useful objective for future work in multivariate conformal prediction.

**Theoretical Claims:**

I did not carefully check the proofs.

---

> ### Author Rebuttal · Authors · 2025-03-31
>
> We thank the reviewer for the detailed reading and the positive comments w.r.t. experiments.
>
> We actually conducted the ones with synthetic data on larger hierarchies (and obtained similar results) but found it difficult to report the results. Figure 2 is already quite complex with the simple 5-2-1 hierarchy. On second thoughts, we could have included a table summarizing the performance on such larger hierarchies (like averages of average coverage and lengths, per layer of the hierarchy). We will do so in the revision.

---

> > ### Comment · Reviewer_4SfW · 2025-04-08
> >
> > Thank you for the reply.

---

### Official Review · Reviewer_zG2b · 2025-03-15

**Overall Recommendation:** 3

**Summary:**

The work proposes application of conformal prediction to hierarchical data. The idea is a combination of two approaches, i.e. split conformal prediction and forecasting reconciliation. The proposed method not only provides global coverage but also component-wise coverage with the computed prediction regions efficient in size.

## update after rebuttal

I think the authors adequately addressed my questions, but I think they could have done a better job at evaluation, hence I will keep my score.

**Claims And Evidence:**

Yes, there is convincing evidence on the efficacy of the method.

**Essential References Not Discussed:**

A key work that discusses Conformal Prediction in Hierarchical setting is missing,

[1] Conformal Prediction in Hierarchical Classification   Thomas Mortier Alireza Javanmardi Yusuf Sale Eyke Hullermeier Willem Waegeman

**Experimental Designs Or Analyses:**

Yes!

**Methods And Evaluation Criteria:**

Yes

**Other Comments Or Suggestions:**

NA

**Other Strengths And Weaknesses:**

The theory seems sound behind the method and the paper is well-written and easy to follow. The experimental setting consisted of 1000 runs on the artificial data, which substantiates the coverage findings, something many conformal works lack.

**Questions For Authors:**

I have a couple of concerns,

1. I don't see a coherent discussion about related works such as [1]
2. The figure 2 is quite busy and could be made more presentable.
3. More datasets could be considered.
4. The baselines are restricted, it might be possible to add other baselines, such as CopulaCPTS as discussed in the work.

**Relation To Broader Scientific Literature:**

Whilst there are not a lot of works considering conformal prediction, there is no coherent discussion on related works.

**Theoretical Claims:**

Yes! Proofs such as coverage guarantees.

---

> ### Author Rebuttal · Authors · 2025-03-31
>
> We thank the reviewer for the detailed reading and the positive comments --- especially the ones relative to the well-conducted and neat experiments.
>
> On the issues raised:
>
> &nbsp;
>
> **1. I don't see a coherent discussion about related works such as [1]**
>
> We will discuss [1] in the revised version. It indeed forms the only other article on conformal prediction that deals with hierarchical data in the way we mean it, but its setting and objectives are much different.
>
> Note: [1] was posted on arXiv on January 31, 2025, the day of the ICML submission deadline.
>
> &nbsp;
>
> **2. The figure 2 is quite busy and could be made more presentable**
>
> This figure is dense, indeed. Since we are providing both graphs in larger sizes in the appendix, we could only keep one in the main body (the one for synthetical data) and for instance, omit the y-axis graduations. This could help getting a more readable picture.
>
> &nbsp;
>
> **3. More datasets could be considered**
>
> Yes, except that creating good base forecasts requires some expertise, which we have as far as electricity load forecasting is concerned, but do not necessarily have for other applications. Due to the nature of our contribution, we focused our submission on the theoretical findings and included a limited number of experiments.
>
> &nbsp;
>
> **4. The baselines are restricted, it might be possible to add other baselines, such as CopulaCPTS as discussed in the work**
>
> Actually, CopulaCPTS (and other methods, which we discuss in the article) target joint coverage (see Equation 2 and Appendix E), instead of the individual coverages stated as the objective (*). This makes these methods not directly comparable. See our answers to Reviewers KKXz and mvP5 for the deep differences between the aims of joint coverage vs. individual coverages.

---

> > ### Comment · Reviewer_zG2b · 2025-04-04
> >
> > Thanks for pointing out the publication date of the work. Indeed, you couldn't mention the work earlier.
> >
> > Creating good base forecasts requires some expertise-- While this is true, the focus of the work is on calibration and not on having good base forecasts, the qualitative assessment can be done easily as long as the base forecaster remains the same, which is typically what most papers on Conformal Prediction do. Also, the work is not a purely theoretical one, it has practical applications, so I disagree on having limited experiments.
> >
> > As for CopulaCPTS, it might not achieve marginal coverages but it could still be compared with. Another work, which I believe provides marginal coverages is:
> >
> > Zhou, Yanfei, Lars Lindemann, and Matteo Sesia. "Conformalized adaptive forecasting of heterogeneous trajectories." Proceedings of the 41st International Conference on Machine Learning. 2024.
> >
> > Just to clear my stance, I am mostly positive about this work, but I am a little disappointed from evaluation perspective.

---

> > > ### Author Response · Authors · 2025-04-07
> > >
> > > **Experiments**
> > >
> > > To be clear, we plan to enrich the experimental section to include real-world i.i.d. datasets (as it also was a concern for Reviewer KKXz). We plan to additionally include larger synthetical hierarchies to illustrate the scalability of our results to complex hierarchies (we already have partial results for this, as discussed in the rebuttal for Reviewer 4SfW). We alas will not be able to show tables or other summaries of results by the end of the discussion phase, but would be able to include such extended results at the time of final ICML publication (which is in several weeks).
> > >
> > > **CopulaCPTS**
> > >
> > > On second thoughts, we could also include experimental results for joint coverage: therein, we would show the impact of the projection step (see Appendix E) and illustrate that we systematically obtain improvements. We could also indeed check how methods targeting joint coverage perform in terms of simultaneous individual coverages but these will be overly conservative, thus much worse in terms of efficiency.
> > >
> > > **On the reference mentioned**
> > >
> > > We checked the mentioned reference by Zhou et al. (2024): what they refer to as "simultaneous marginal coverage" is not the component-wise coverages we are targeting in our submission (that we also refer to as individual coverages in this review thread); see their Equation (1) and the paragraph above: they still target joint coverages, but simultaneously over several data points (simultaneously over time).
> > >
> > > **As a conclusion**
> > >
> > > We are pleased that you appreciate the theoretical part of our work and the experimental study already included. All reviewers of this thread seem to do so, while asking for an extended section of experimental results. This is something we commit to perform (and the submitted work should prove that this commitment is feasible).

---

### Decision · Program_Chairs · 2025-05-01

**Decision:**

Reject

**Comment:**

This study tackles the problem of conformal prediction in the context of data with hierarchical structure. The authors claimed that incorporating hierarchical information can yield tighter prediction sets compared to standard conformal prediction methods. The paper received both positive and negative evaluations from reviewers. While all reviewers agreed that the problem is important and the proposed approach is both interesting and potentially effective. However, some reviewers with expertise in conformal prediction and related areas still have concerns, even after the rebuttal, about the lack of sufficient experimental and theoretical detail, which makes it difficult to assess the validity of the method. While this is an interesting and potentially important contribution to the machine learning and statistics communities, the current manuscript requires substantial revisions to address these issues and is not suitable for acceptance in its current form.